# Asymptotically Optimal Exact Minibatch Metropolis-Hastings

**Ruqi Zhang**[*]
Cornell University
rz297@cornell.edu

**A. Feder Cooper**[*]
Cornell University
afc78@cornell.edu

**Christopher De Sa**
Cornell University
cdesa@cs.cornell.edu

## Abstract

Metropolis-Hastings (MH) is a commonly-used MCMC algorithm, but it can be intractable on large datasets due to requiring computations over the whole dataset. In this paper, we study *minibatch MH* methods, which instead use subsamples to enable scaling. We observe that most existing minibatch MH methods are inexact (i.e. they may change the target distribution), and show that this inexactness can cause arbitrarily large errors in inference. We propose a new exact minibatch MH method, *TunaMH*, which exposes a tunable trade-off between its batch size and its theoretically guaranteed convergence rate. We prove a lower bound on the batch size that any minibatch MH method *must* use to retain exactness while guaranteeing fast convergence—the first such bound for minibatch MH—and show TunaMH is asymptotically optimal in terms of the batch size. Empirically, we show TunaMH outperforms other exact minibatch MH methods on robust linear regression, truncated Gaussian mixtures, and logistic regression.

## 1 Introduction

Bayesian inference is widely used for probabilistic modeling of data. Specifically, given a dataset $\mathcal{D} = \{x_i\}_{i=1}^N$ and a $\theta$-parameterized model, it aims to compute the posterior distribution

$$\pi(\theta) \propto \exp\left(-\sum_{i=1}^N U_i(\theta)\right), \text{ where } U_i(\theta) = -\log p(x_i|\theta) - \frac{1}{N}\log p(\theta).$$

Here $p(\theta)$ is the prior and the $p(x_i|\theta)$ give the likelihood of observing $x_i$ given the parameter $\theta$. We assume the data are conditionally independent given $\theta$. The $U_i$ have a natural interpretation as component *energy functions* with $\pi$ acting as a Gibbs measure. In practice, computing $\pi(\theta)$ is often intractable and thus requires using approximate methods, such as Markov chain Monte Carlo (MCMC). MCMC uses sampling to estimate the posterior and is guaranteed to converge asymptotically to the true distribution, $\pi$ [9].

The Metropolis-Hastings (MH) algorithm [16, 21] is one of the most commonly used MCMC methods. In each step, MH generates a proposal $\theta'$ from a distribution $q(\cdot|\theta)$, and accepts it with probability

$$a(\theta, \theta') = \min\left(1, \frac{\pi(\theta')q(\theta|\theta')}{\pi(\theta)q(\theta'|\theta)}\right) = \min\left(1, \exp\left(\sum_{i=1}^N (U_i(\theta) - U_i(\theta'))\right) \cdot \frac{q(\theta|\theta')}{q(\theta'|\theta)}\right). \quad (1)$$

If accepted, the chain transitions to $\theta'$; otherwise, it remains at the current state $\theta$. This accept/reject step can be quite costly when $N$ is large, since it entails computing a sum over the entire dataset.

Prior work has proposed many approaches to mitigate the cost of this decision step [5]. One popular approach involves introducing stochasticity: instead of computing over the entire dataset, a subsample, or *minibatch*, is used to compute an approximation. These minibatch MH methods can be divided into

---

[*]Equal contribution.

two classes, *exact* and *inexact*, depending on whether or not the target distribution $\pi$ is necessarily preserved. Inexact methods introduce asymptotic bias to the target distribution, trading off correctness for speedups [6, 17, 23, 24, 26]. Exact methods either require impractically strong constraints on the target distribution [20, 27], limiting their applicability in practice, or they negatively impact efficiency, counteracting the speedups that minibatching aims to provide in the first place [4, 12]. Moreover, all existing exact methods operate on the belief that there is a trade-off between batch size and convergence rate—between scalability and efficiency. Yet no prior work formally exposes this trade-off, and most prior work gives no convergence rate guarantees. Given these various considerations, it is not entirely clear how to evaluate which minibatch MH method to use.

In this paper we forge a path ahead to untangle this question. While inexact methods have been prominent recently due to their efficiency, they are not reliable: we show that the stationary distribution of any inexact method can be arbitrarily far from the target $\pi$. This means they can yield disastrously wrong inference results in practice, and it is difficult to tell just how bad those results can be.

We therefore turn our attention to exact methods and introduce *TunaMH*.[2] Compared to prior work, we make milder assumptions, which enables TunaMH to apply to a wider variety of inference tasks. More specifically, we require local rather than global bounds on the target distribution [20, 27] and do not rely on the Bernstein-von Mises approximation [5, 7, 12]. TunaMH is guaranteed to retain sample efficiency in the presence of minibatching: its convergence rate (measured by the spectral gap) is within a constant factor of standard, non-minibatch MH. More importantly, TunaMH also enables us to rigorously characterize the trade-off between scalability and efficiency. It has a hyperparameter $\chi$, which enables tuning the trade-off between expected batch size and convergence rate.

By exposing this trade-off, our analysis raises the natural question: *is TunaMH optimal for this trade-off?* That is, could another exact algorithm use an asymptotically smaller average batch size while having the same convergence rate guarantees? We explore this in Section 4; under the same mild assumptions we use to derive TunaMH, we prove a lower bound on the expected batch size for *any* exact minibatch MH method that can keep a reasonable convergence rate. To our knowledge, we are the first to prove a lower bound of this nature for minibatch MH. Moreover, TunaMH is *asymptotically optimal* in balancing the expected batch size and convergence rate. It remains exact and efficient while on average using the smallest possible number of samples. In summary:

- We demonstrate that any inexact minibatch MH method can be arbitrarily inaccurate (Section 2.1).
- We introduce a new exact method, TunaMH (Section 3), with a lower bound on its convergence rate (in terms of the spectral gap) and a tunable hyperparameter to balance the trade-off between convergence rate and batch size.
- We prove a lower bound on the batch size for any exact minibatch MH method given a target convergence rate—the first such lower bound in this area. This result indicates that the expected batch size of TunaMH is asymptotically optimal in terms of the problem parameters (Section 4).
- We show empirically that TunaMH outperforms state-of-the-art exact minibatch MH methods on robust linear regression, truncated Gaussian mixture, and logistic regression (Section 5).

## 2 Preliminaries and Drawbacks of Prior Minibatch MH Methods

We first formally define the class of methods that we study theoretically in this paper: minibatch MH methods of the form of Algorithm 1. This class contains methods that sample a proposal from distribution $q$ (which we always assume results in the chain being ergodic), and choose to accept or reject it by calling some randomized subroutine, `SubsMH`, which outputs $1$ or $0$ for "accept" or "reject," respectively. Algorithms in this class have several notable properties. First, `SubsMH` is *stateless*: each acceptance decision is made independently, without carrying over local state associated with the MH procedure between steps. Many prior methods are stateless [6, 12, 17, 26]. We do not consider *stateful* methods, in which the decision depends on previous state; they are difficult to analyze due to running on an extended state space [3, 24]. Second, `SubsMH` takes a function that computes energy *differences* $U_i(\theta) - U_i(\theta')$ and outputs an acceptance decision. We evaluate efficiency in terms of how many times `SubsMH` calls this function, which we term the *batch size* the method uses. Third, `SubsMH` takes parameters that bound the maximum magnitude of the energy differences. Specifically, as in Cornish et al. [12], we assume:

**Algorithm 1** Stateless, Energy-Difference-Based Minibatch Metropolis-Hastings

---

**given:** state space $\Theta$, energy functions $U_1, \ldots, U_N : \Theta \to \mathbb{R}$, proposal dist. $q$, initial state $\theta \in \Theta$
**given:** parameters $c_1, \ldots, c_N$, $C$, $M$ from Assumption 1, randomized algorithm `SubsMH`
**loop**
    **sample** $\theta' \sim q(\cdot | \theta)$
    **define function** $\Delta U : \{1, \ldots, N\} \to \mathbb{R}$, such that $\Delta U(i) = U_i(\theta) - U_i(\theta')$
    **call subroutine** $o \leftarrow$ `SubsMH`$(\Delta U, N, q(\theta | \theta')/q(\theta' | \theta), c_1, \ldots, c_N, C, M(\theta, \theta'))$
    **if** $o = 1$, **update** $\theta \leftarrow \theta'$
**end loop**

---

**Assumption 1.** *For some constants $c_1, \ldots, c_N \in \mathbb{R}_+$, with $\sum_i c_i = C$, and symmetric function $M :$ $\Theta \times \Theta \to \mathbb{R}_+$, for any $\theta, \theta' \in \Theta$, the energy difference is bounded by $|U_i(\theta) - U_i(\theta')| \leq c_i M(\theta, \theta')$.*

One can derive such a bound, which can be computed in $O(1)$ time, for many common inference problems: for example, if each energy function $U_i$ is $L_i$-Lipschitz continuous, then it suffices to set $c_i = L_i$ and $M(\theta, \theta') = \|\theta - \theta'\|$ (See Appendix J for examples of $c_i$ and $M$ on common problems). Note that the `SubsMH` method may choose *not* to use these bounds in its decision. We allow this so the form of Algorithm 1 can include methods that do not require such bounds. Most existing methods can be described in this form [4, 6, 12, 17, 26]. For example, standard MH can be written by setting `SubsMH` to a subroutine that computes the acceptance rate $a$ as in (1) and outputs 1 (i.e., accept) with probability $a$.

Such minibatch MH methods broadly come in two flavors: *inexact* and *exact*. We next establish the importance of being exact and demonstrate how TunaMH resolves drawbacks in prior work.

## 2.1 The Importance of Being Exact

Inexact methods are popular due to helping scale MH to new heights [6, 17, 24, 26]. They approximate the MH acceptance ratio to within an error tolerance ($> 0$), trading off exactness for efficiency gains. Surprisingly, the bias from inexactness can be arbitrarily large even when the error tolerance is small.

**Theorem 1.** *Consider any minibatch MH method of the form in Algorithm 1 that is inexact (i.e. does not necessarily have $\pi$ as its stationary distribution for all $\pi$ satisfying Assump. 1). For any constants $\delta \in (0, 1)$ and $\rho > 0$, there exists a target distribution $\pi$ and proposal distribution $q$ such that if we let $\tilde{\pi}$ denote a stationary distribution of the inexact minibatch MH method on this target, it satisfies*

$$\mathrm{TV}(\pi, \tilde{\pi}) \geq \delta \text{ and } \mathrm{KL}(\pi, \tilde{\pi}) \geq \rho.$$

*where TV is the total variation distance and KL is the Kullback–Leibler divergence.*

Theorem 1 shows that when using any inexact method, there always exists a target distribution $\pi$ (factored in terms of energy functions $U_i$) and proposal distribution $q$ such that it will approximate $\pi$ arbitrarily poorly. This can happen even when individual errors are small; they can still accumulate a very large overall error. We prove Theorem 1 via a simple example—a random walk along a line, in which the inexact method causes the chain to step towards one direction more often than the other, even though its steps should be balanced (Appendix A). Note that it may be possible to avoid a large error by using some specific proposal distribution, but such a proposal is hard to know in general.

We use AustereMH [17] and MHminibatch [26] to empirically validate Theorem 1. For these inexact methods, we plot density estimates with the number of states $K = 200$ in Figure 1a (see Appendix J.1 for using other $K$); the stationary distribution diverges from the target distribution significantly. Moreover, the TV distance between the density estimate and the true density increases as $K$ increases on this random walk example (Figure 1b). By contrast, our exact method (Section 3) keeps a small TV distance on all $K$ and estimates the density accurately with an even smaller average batch size. We also tested AustereMH on robust linear regression, a common task, to show that the error of inexact methods can be large on standard problems (Appendix J.1).

## 2.2 Issues with Existing Exact Methods

This observation suggests that we should be using exact methods when doing minibatch MH. However, existing approaches present additional drawbacks, which we discuss below.

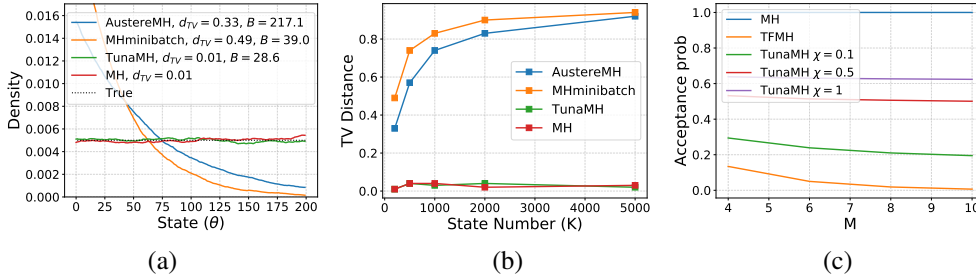

Figure 1: Existing MH method issues. (a)-(b) Inexact methods can diverge a lot from true distribution. "$d_{TV}$" and "$B$" denote the TV distance and the batch size respectively. (c) SMH has low and TunaMH with different values of hyperparameter $\chi$ has high acceptance rates.

**Factorized MH and Scalable MH** are stateless, exact minibatch methods. Factorized MH (FMH) decomposes the acceptance rate into a product of factors, which allows for rejecting a proposal based on a minibatch of data [4, 10, 11]. Truncated FMH (TFMH) is a FMH variant that maintains geometric ergodicity; it falls back on standard MH in a step when the bound on the factors reaches a certain threshold [12]. No matter how this threshold is set, we can construct tasks where TFMH is either arbitrarily inefficient (rejecting arbitrarily often, slowing convergence), or degrades entirely to standard MH.

**Statement 1.** *For any constant $p \in (0, 1)$, there exists a target distribution such that TFMH either has an acceptance rate which is less than p times that of standard MH, or it completely degrades to standard MH (summing over the whole dataset at each step).*

We prove this statement in Appendix C using an example of a uniform distribution along a line, where we let $x_i$ take one of two values, $\{-M/N, M/N\}$ with $M > 0$. We show that the acceptance rate of TFMH can be arbitrarily low by increasing $M$, which we also empirically verify in Figure 1c.

To improve the acceptance rate of TFMH, Scalable MH (SMH) introduces control variates, which approximate $U_i$ with a Taylor series around the mode [12]. However, it only works with unimodal posteriors and high-quality Bernstein-von Mises approximations—conditions that do not hold for many common inference tasks.

**PoissonMH** is a stateless minibatch MH method adapted from an algorithm designed for scaling Gibbs sampling on factor graphs [27]. However, unlike our method, it requires strong assumptions—specifically, a global upper bound on the energy. Such an upper bound usually does not exist and, even if it does, can be very large, resulting in an impractically large batch size.

**FlyMC** is a stateful method, which means it uses auxiliary random variables to persist state across different MH steps [20]. It requires a lower bound on the likelihood function, which is typically more demanding than Assumption 1 and does not have theoretical performance guarantees.

**Other exact methods** exist based on Piecewise Deterministic Markov Processes [7, 8]. They require regularity conditions only available for some problems, so their practical utility is limited.

## 3 TunaMH: Asymptotically Optimal Exact MH

In this section, we present our method, TunaMH, which evades the issues of prior exact methods discussed in Section 2.2. Like SMH [12], our method works on distributions for which an *a priori* bound on the energy differences is known (Assumption 1).

Our algorithm, presented in Algorithm 2, takes as parameters $c_1, \ldots, c_N, C$, and $M$ from Assumption 1, along with an additional hyperparameter, $\chi > 0$. It proceeds in four steps. First, like any MH method, it generates a proposal $\theta'$ from given distribution $q$. Second, it samples a batch size $B$ from a Poisson distribution. This makes the expected number of energy functions $U_i$ evaluated by our method at each step $\mathbf{E}[B] = \chi C^2 M^2(\theta, \theta') + CM(\theta, \theta')^3$. Importantly, this means the batch

---
<sup>3</sup>Note that $\mathbf{E}[B]$ is typically $<< N$ and can be decreased using small step sizes. If, however, $\mathbf{E}[B] > N$, then we can simply use standard MH in that iteration, similar to TFMH.

---
**Algorithm 2** TunaMH
---

**given:** initial state $\theta \in \Theta$; proposal dist. $q$; hyperparameter $\chi$; Asm. 1 parameters $c_i, C, M$
**loop**
    **propose** $\theta' \sim q(\cdot|\theta)$ and **compute** $M(\theta, \theta')$
    ▷ Form minibatch $\mathcal{I}$
    **sample** $B \sim \text{Poisson}\left(\chi C^2 M^2(\theta, \theta') + CM(\theta, \theta')\right)$
    **initialize minibatch indices** $\mathcal{I} \leftarrow \emptyset$ (an initially empty multiset)
    **for** $b \in \{1, \ldots, B\}$ **do**
        **sample** $i_b$ such that $\mathbf{P}(i_b = i) = c_i/C$, for $i = 1 \ldots N$
        **with probability** $\frac{\chi c_{i_b} CM^2(\theta,\theta') + \frac{1}{2}(U_{i_b}(\theta') - U_{i_b}(\theta) + c_{i_b} M(\theta,\theta'))}{\chi c_{i_b} CM^2(\theta,\theta') + c_{i_b} M(\theta,\theta')}$ **add** $i_b$ to $\mathcal{I}$
    **end for**
    ▷ Accept/reject step based on minibatch $\mathcal{I}$
    **compute MH ratio** $r \leftarrow \exp\left(2 \sum_{i \in \mathcal{I}} \text{artanh}\left(\frac{U_i(\theta) - U_i(\theta')}{c_i M(\theta,\theta')(1 + 2\chi CM(\theta,\theta'))}\right)\right) \cdot \frac{q(\theta'|\theta)}{q(\theta|\theta')}$
    **with probability** $\min(1, r)$, set $\theta \leftarrow \theta'$
**end loop**

---

size may vary from iteration to iteration, and the expected size depends on $\theta$ and $\theta'$. For example, TunaMH may tend to set $B$ larger for larger-distance proposals with a higher $M(\theta, \theta')$. Third, it samples (with replacement) a minibatch of size $B$, but for each data point it samples, it has some probability of *ejecting* this point from the minibatch. Finally, it accepts the proposed $\theta'$ with some probability, computed using a sum over the post-ejection minibatch. Our method can be derived by carefully replacing the auxiliary variables in PoissonMH with *local* Poisson variables whose distributions change each iteration depending on the pair $(\theta, \theta')$ (Appendix D). By construction TunaMH is exact; it preserves the target distribution $\pi$ as its stationary distribution. This is because TunaMH is *reversible*, meaning its transition operator $T$ satisfies $\pi(\theta)T(\theta, \theta') = \pi(\theta')T(\theta', \theta)$ for any $\theta, \theta' \in \Theta$. This is a common condition that guarantees that a MCMC method has $\pi$ as its stationary distribution [9, 18].

Compared to previous exact methods, a significant benefit of TunaMH is that we can prove theoretical guarantees on its efficiency. Specifically, its convergence speed is guaranteed to be close to standard MH and $\chi$ allows us to control how close. To show this, we lower bound the convergence rate of TunaMH in terms of the *spectral gap*, which is commonly used to characterize convergence speed in the MCMC literature [15, 18, 25, 27, 28]. The larger the spectral gap, the faster the chain converges.

**Definition 1.** *The* spectral gap *of a reversible Markov chain is the distance between the largest and second-largest eigenvalues of its transition operator. That is, if the eigenvalues of the transition operator are $1 = \lambda_1 > \lambda_2 \geq \lambda_3 \cdots$, then the spectral gap is $\gamma = 1 - \lambda_2$.*

**Theorem 2.** *TunaMH (Algorithm 2) is reversible with stationary distribution $\pi$. Let $\bar{\gamma}$ denote the spectral gap of TunaMH, and let $\gamma$ denote the spectral gap of standard MH with the same target distribution and proposal distribution. Then,*

$$\bar{\gamma} \geq \exp\left(-\tfrac{1}{\chi} - 2\sqrt{\tfrac{\log 2}{\chi}}\right) \cdot \gamma.$$

Intuitively, this theorem (proof in Appendix E) suggests the convergence rate of TunaMH is at most a constant slower than that of standard MH, and can be increased by adjusting the hyperparameter $\chi$. Recall that $\chi$ also controls the batch size of TunaMH. Effectively, this means $\chi$ is a *dial* that allows us to directly tune the trade-off between convergence rate and batch size. When $\chi$ is large, the batch size $B$ is large and the spectral gap ratio, $\bar{\gamma}/\gamma$, is close to 1: the larger batch size is less scalable but keeps a high convergence rate. Conversely, when $\chi$ is small, the batch size is small and the spectral gap ratio is close to 0: we trade off slow-downs in convergence rate for scalability. For example, for any $0 < \kappa < 1$, to guarantee the spectral gap ratio $\bar{\gamma}/\gamma \geq \kappa$ it suffices to set (Appendix F)

$$\chi = \tfrac{4}{(1-\kappa)\log(1/\kappa)}, \quad \text{giving an average batch size of } \quad \mathbf{E}[B] = \tfrac{4C^2 M^2(\theta,\theta')}{(1-\kappa)\log(1/\kappa)} + CM(\theta, \theta'). \quad (2)$$

In practice, we usually want to minimize the wall-clock time to achieve a certain estimate error, which requires tuning $\chi$ to optimally balance scalability and efficiency. We attempt to derive a theoretically

optimal value of $\chi$ in Appendix G by minimizing the product of the relaxation time—a measure of the number of steps needed—and the expected wall-clock time per step. Note that this product may be loose in bounding the total wall-clock time (we leave tightening this bound to future work), making the derived $\chi$ larger than necessary. In Section 5 we give a simple heuristic to tune $\chi$, which works well and is generally better than the derived value.

Theorem 2 only requires the mild constraints of Assumption 1 on the target distribution, so applies in many scenarios and compares well to other exact methods. SMH further requires a Bernstein-von Mises approximation to have guarantees on its batch size and acceptance rate. PoissonMH provides convergence rate guarantees, but demands the strong assumption that the target distribution has a global upper bound on the energy. FlyMC does not have any theoretical guarantees on performance.

## 4 Towards Optimal Exact Minibatch MH

In Theorem 2, we expose the trade-off between convergence rate and batch size in TunaMH. Here, we take this analysis a step further to investigate the limits of how efficient an exact minibatch MH method can be. To tackle this problem, we derive a lower bound on the batch size for any minibatch MH method that retains exactness and fast convergence. We then show that TunaMH is asymptotically optimal in terms of its dependence on the problem parameters $C$ and $M$. In other words, it is not possible to outperform TunaMH in this sense with a method in the class described by Algorithm 1.

**Theorem 3.** *Consider any stateless exact minibatch MH algorithm described by Algorithm 1, any state space $\Theta$ (with $|\Theta| \geq 2$), any $C > 0$, and any function $M : \Theta \times \Theta \to \mathbb{R}^+$. Suppose that the algorithm guarantees that, for some constant $\kappa \in (0, 1)$, for any distribution, the ratio between the spectral gap of minibatch MH $\hat{\gamma}$ and the spectral gap of standard MH $\gamma$ is bounded by $\hat{\gamma} \geq \kappa\gamma$. Then there must exist a distribution $\pi$ over $\Theta$ and proposal $q$ such that the batch size $B$ of that algorithm, when deciding whether to accept any transition $\theta \to \theta'$, is bounded from below by*

$$\mathbf{E}[B] \geq \zeta \cdot \kappa \cdot \left( C^2 M^2(\theta, \theta') + CM(\theta, \theta') \right) \tag{3}$$

*for some constant $\zeta > 0$ independent of algorithm and problem parameters.*

To prove this theorem, we construct a random walk example over two states, then consider the smallest batch size a method requires to distinguish between two different stationary distributions (Appendix H). The impact of Theorem 3 is three-fold:

First, it provides an upper bound on the performance of algorithms of Algorithm 1's form: in each iteration, the average batch size of any exact minibatch MH method of the form of Algorithm 1 must be set as in (3) in order to maintain a reasonable convergence rate. To the best of our knowledge, this is the first theorem that rigorously proves a ceiling for the possible performance of minibatch MH.

Second, TunaMH achieves this upper bound. In fact, Theorem 3 suggests that TunaMH is *asymptotically optimal* in terms of the problem parameters, $C$ and $M$. To see this, observe that when we ignore $\kappa$, both expressions that bound $\mathbf{E}[B]$ in (2) and (3) are $\mathcal{O}(C^2 M^2(\theta, \theta') + CM(\theta, \theta'))$. Thus TunaMH reaches the lower bound, achieving asymptotic optimality in terms of $C$ and $M$. (Of course, this sense of "optimality" does not rule out potential constant-factor improvements over TunaMH or improvements that depend on $\kappa$.)

Lastly, this result suggests directions for developing new exact minibatch MH algorithms: to be significantly faster than TunaMH, we either need to introduce additional assumptions to the problem or to develop new stateful algorithms.

In prior work, when assuming a very concentrated posterior, some methods' batch size can scale in $\mathcal{O}(1)$ [5, 7, 12] or $\mathcal{O}(1/\sqrt{N})$ [12] in terms of the dataset size $N$ while maintaining efficiency. Theorem 3 is compatible with these results, further demonstrating this is essentially the *best* dependency on $N$ an exact minibatch MH method can achieve. We show this by explicitly assuming the dependency of $C$ and $M$ on $N$, as in SMH [12], yielding the following corollary (proof in Appendix I):

**Corollary 1.** *Suppose that $C$ increases linearly with $N$ ($C = \mathcal{O}(N)$) and $M(\theta, \theta')$ scales in $\mathcal{O}(N^{-(h+1)/2})$ for some constant $h > 0$. Then the lower bound in Theorem 3 becomes $\mathcal{O}(N^{(1-h)/2})$. In particular, it is $\mathcal{O}(1)$ when $h = 1$, and $\mathcal{O}(1/\sqrt{N})$ when $h = 2$.*

That is, TunaMH matches the state-of-the-art's dependency on $N$, and this dependency is optimal. Similarly, since $C$ and $M$ are the only problem parameters in the lower bound in Theorem 3, we can

also get the optimal dependency on the other problem parameters by explicitly assuming the relation of them with $C$ and $M$.

# 5 Experiments

We compare TunaMH to MH, TFMH, SMH (i.e. TFMH with MAP control variates) and FlyMC. We only include PoissonMH in the Gaussian mixture experiment, as it is not applicable in the other tasks. All of these methods are unbiased, so they have the same stationary distribution. To ensure fair wall-clock time comparisons, we coded each method in Julia; our implementations are at least as fast as, if not faster than, prior implementations. For each trial, we use Gaussian random walk proposals. We tune the proposal stepsize separately for each method to reach a target acceptance rate, and report averaged results and standard error from the mean over three runs. We set $\chi$ to be roughly the largest value that keeps $\chi C^2 M^2(\theta, \theta') < 1$ in most steps; we keep $\chi$ as high as possible while the average batch size is around its lower bound $CM(\theta, \theta')$. We found this strategy works well in practice. We released the code at `https://github.com/ruqizhang/tunamh`.

## 5.1 Robust Linear Regression

We first test TunaMH on robust linear regression [12, 20]. We use a Student's t-distribution with degree of freedom $v = 4$ and set data dimension $d = 100$ (Appendix J). We tune each method separately to a 0.25 target acceptance rate. To measure efficiency, we record effective sample size (ESS) per second—a common MCMC metric for quantifying the number of effectively independent samples a method can draw from the posterior each second [9]. Figure 2a shows TunaMH is the most efficient for all dataset sizes $N$; it has the largest ESS/second. For minibatch MH methods, Figure 2b compares the average batch size. TunaMH's batch size is significantly smaller than FlyMC's—about 35x with $N = 10^5$. TFMH has the smallest batch size, but this is because it uses a very small step size to reach the target acceptance rate (Table 2 in Appendix J.2). This leads to poor efficiency, which we can observe in its low ESS/second.

**MAP variants** Since TFMH and FlyMC have variants that use the *maximum a posteriori* (MAP) solution to boost performance, we also test TunaMH in this scheme. SMH uses MAP to construct control variates for TFMH to improve low acceptance rates. We consider both first- and second-order approximations (SMH-1 and SMH-2). FlyMC uses MAP to tighten the lower bound (FlyMC-MAP). For our method (TunaMH-MAP) and MH (MH-MAP), we simply initialize the chain with the MAP solution. Figure 2c shows that TunaMH performs the best even when previous methods make use of MAP. With control variates, SMH does increase the acceptance rate of TFMH, but this comes at the cost of a drastically increased batch size (Figure 2d) which we conjecture is due to the control variates scaling poorly in high dimensions ($d = 100$).[4] FlyMC-MAP tightens the bounds, entailing a decrease in the batch size. However, as clear in the difference in ESS/second, it is still less efficient than TunaMH due to its strong dependence between auxiliary variables and the model parameters—an issue that previous work also documents [24].

## 5.2 Truncated Gaussian Mixture

Next we test on a task with a multimodal posterior, a very common problem in machine learning. This demonstrates the advantage of TunaMH not relying on MAP, because MAP is a single solution and therefore is unable to reflect all possible modes in multimodal distributions. As a result, methods that rely on MAP tuning or MAP-based control variates are unable to perform well on such problems.

We consider a Gaussian mixture. To get bounds on TunaMH, TFMH, SMH, and FlyMC, we truncate the posterior, bounding $\theta_1, \theta_2 \in [-3, 3]$ similar to Zhang and De Sa [27]. We can include PoissonMH because its required bound exists after truncation. As in Seita et al. [26], we use a tempered posterior $\pi(\theta) \propto \exp\left(-\beta \sum_i U_i(\theta)\right)$ with $N = 10^6$ and $\beta = 10^{-4}$. Figure 3a compares performance, showing symmetric KL versus wall-clock time. TunaMH is the fastest, converging after 1 second, whereas the others take much longer. As expected, SMH-1 performs worse than TFMH, verifying the control variate is unhelpful for multimodal distributions. FlyMC and FlyMC-MAP are also inefficient; their performance is on par with standard MH, indicating negligible benefits from minibatching.

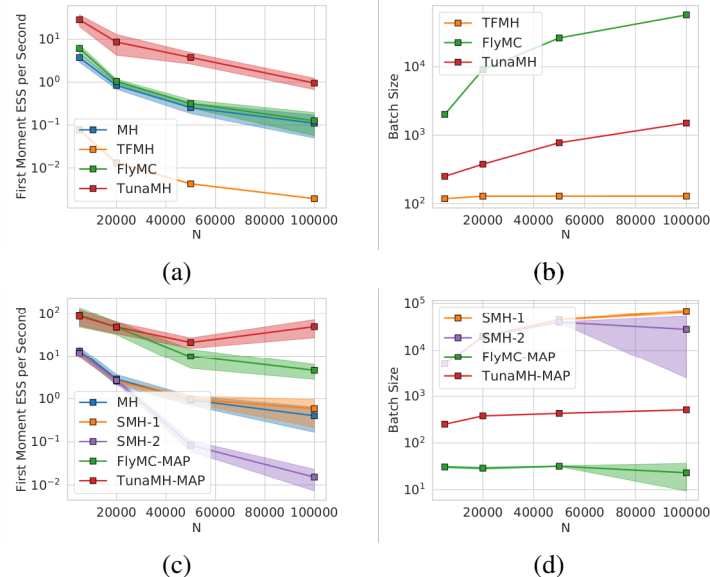

(a)                 (b)

(c)                 (d)

Figure 2: Robust linear regression, $d = 100$. (a) ESS/second without MAP. (b) Average batch size without MAP. (c) ESS/second with MAP. (d) Average batch size with MAP.

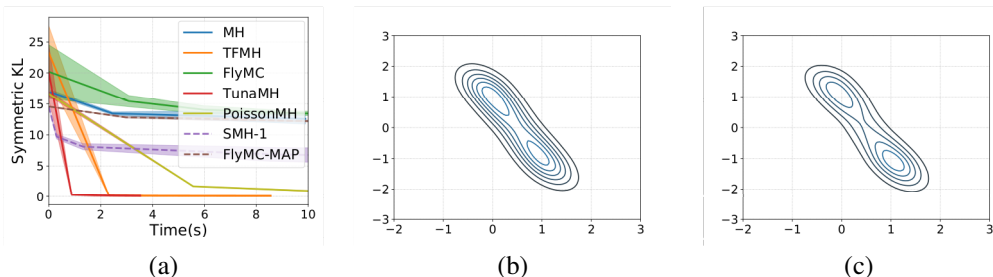

(a)                 (b)                 (c)

Figure 3: Truncated Gaussian mixture. (a) Symmetric KL comparison. (b) True distribution. (c) Denstity estimate of TunaMH after 1 second.

TunaMH also performs significantly better in terms of batch size, especially in comparison to PoissonMH (Table 1). This is due to TunaMH's local bound on the energy, as opposed to PoissonMH's global bound. This also allows TunaMH to run on more problem types, such as robust linear (Section 5.1) and logistic (Section 5.3) regression. To illustrate the estimate quality, we also visualize the density estimate after 1 second; TunaMH's estimate (Figure 3c) is very close to the true distribution (Figure 3b), while the other methods do not provide on-par estimates within the same time budget (Appendix J.3).

### 5.3 Logistic Regression on MNIST

Lastly we apply TunaMH to logistic regression on the MNIST image dataset of handwritten number digits. Mirroring the work of FlyMC [20], we aim to classify 7s and 9s using the first 50 principal components as features. We set $\chi = 10^{-5}$ following our heuristic. In Figure 4a we see that TunaMH is the fastest of all methods to converge, as measured by wall-clock time. We also compare average batch size in Table 1. TunaMH's average batch size is 4x smaller than FlyMC's. TFMH again has the smallest batch size, but sacrifices efficiency by using a small step size in order to achieve the target acceptance rate. Thus, overall, TFMH is again inefficient in these experiments.

**Effect of Hyperparameter $\chi$** To understand the effect of $\chi$ in TunaMH, we report results with varying $\chi$. Figure 4b plots test accuracy as a function of the number of iterations. As $\chi$ increases, TunaMH's convergence rate approaches standard MH. This verifies our theoretical work: $\chi$ acts like a dial to control convergence rate and batch size trade-off—mapping to the efficiency-scalability

Table 1: Avg. batch size $\pm$ SE from the mean on 3 runs. PoissonMH not applicable to logistic reg.

| Tasks | TFMH | FlyMC | PoissonMH | TunaMH |
|---|---|---|---|---|
| Gaussian Mixture | $13.91 \pm 0.016$ | $811.52 \pm 234.16$ | $3969.67 \pm 327.26$ | $86.45 \pm 0.04$ |
| Logistic Regression | $39.28 \pm 0.12$ | $1960.19 \pm 150.96$ | — | $504.07 \pm 0.33$ |

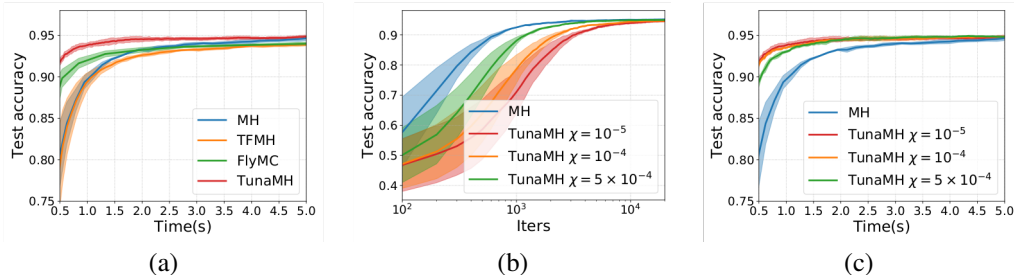

| (a) | (b) | (c) |
|---|---|---|

Figure 4: MNIST logistic regression. (a) Test accuracy comparison. (b)-(c) TunaMH's test accuracy for various $\chi$. Batch size for $\chi = 10^{-5}, 10^{-4}, 5 \times 10^{-4}$ is 504.07, 810.35 and 2047.91 respectively.

trade-off. Figure 4c shows TunaMH's wall-clock time performance is not sensitive to $\chi$, as the performance is superior to standard MH regardless of how we set it. However, $\chi$ needs to be tuned in order to achieve the best performance. Previous methods do not have such a dial, so they are unable to control this trade-off to improve the sampling efficiency.

## 6  Conclusion and Future Work

After demonstrating that inexact methods can lead to arbitrarily incorrect inference, we focus our work in this paper on exact minibatch MH methods. We propose a new exact method, TunaMH, which lets users trade off between batch size and guaranteed convergence rate—between scalability and efficiency. We prove a lower bound on the batch size that any minibatch MH method must use to maintain exactness and convergence rate, and show TunaMH is asymptotically optimal. Our experiments validate these results, demonstrating that TunaMH outperforms state-of-the-art exact methods, particularly on high-dimensional and multimodal distributions.

To guide our analysis, we formalized a class of stateless, energy-difference-based minibatch MH methods, to which most prior methods belong. While TunaMH is asymptotically optimal for this class, future work could develop new exact methods that are better by a constant factor or on some restricted class of distributions. It would also be interesting to develop effective theoretical tools for analyzing stateful methods, since these methods could potentially bypass our lower bound.

## Broader Impact

Our work shines a light on how to scale MCMC methods responsibly. We make the case that inexact minibatch MH methods can lead to egregious errors in inference, which suggests that—particularly for high-impact applications [14, 22]—we should avoid their use. We provide an alternative: a minibatch MH method that guarantees correctness, while also maintaining an optimal balance between efficiency and scalability, enabling its safe use on large-scale applications.

## Acknowledgements

This work was supported by a gift from SambaNova Systems, Inc. and funding from Adrian Sampson. We thank Jerry Chee, Yingzhen Li, and Wing Wong for helpful feedback on the manuscript.

## Footnotes

[2]TunaMH since it *tunes* the efficiency-scalability trade-off and uses a Poisson (French for "fish") variable.

[4]Control variates worked well in the SMH paper [12] because all experiments had small dimension ($d = 10$).

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
