[Supplementary Material]

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

# A  Proof of Theorem 1

In this section, we prove Theorem 1, which asserts that any inexact stateless MH algorithm can produce arbitrarily large bias between its target distribution (the distribution we are trying to sample from) and its stationary distribution (the distribution that the chain actually produces samples from asymptotically).

*Proof.* Let $\mathcal{A}$ denote the SubsMH in Algorithm 1 of the minibatch MH method in question. Since $\mathcal{A}$ is inexact, there must exist a state space $\Theta$, proposal distribution $q$, and target distribution $\mu$, satisfying Assumption 1 with parameters $c_1, \ldots, c_N, C, M$, where

$$\mu(\theta) \propto \exp\left( -\sum_{i=1}^{N} V_i(\theta) \right)$$

for some $N$ and energy functions $V_1, \ldots, V_N$, such that $\mathcal{A}$ run on $\mu$ with proposal distribution $q$ does not have stationary distribution $\mu$.

Next, let $a_\mu(\theta, \theta')$ denote the acceptance probability of algorithm $\mathcal{A}$ on the above task for a proposed transition from $\theta$ to $\theta'$. Assume by way of contradiction that on this problem, it is always true that

$$\frac{a_\mu(\theta, \theta')}{a_\mu(\theta', \theta)} = \frac{\mu(\theta')q(\theta|\theta')}{\mu(\theta)q(\theta'|\theta)}.$$

If this were true, then the overall transition probability of this chain, for $\theta \neq \theta'$, would be

$$T_\mu(\theta, \theta') = q(\theta'|\theta) \cdot a_\mu(\theta, \theta')$$

and it would hold that

$$\mu(\theta)T_\mu(\theta, \theta') = \mu(\theta')T_\mu(\theta', \theta).$$

That is, the chain would be reversible, also known as satisfying detailed balance. But it is a standard result that for any reversible chain, $\mu$ must be a stationary distribution of that chain. We have now derived a contradiction, which establishes that our assumption is false. That is, there exists a $\theta, \theta' \in \Theta$ such that

$$\frac{a_\mu(\theta, \theta')}{a_\mu(\theta', \theta)} \neq \frac{\mu(\theta') \cdot q(\theta|\theta')}{\mu(\theta) \cdot q(\theta'|\theta)}.$$

Explicitly, this means that if we define the function $\Delta V$ such that

$$\Delta V(i) = V_i(\theta) - V_i(\theta'),$$

then for this subsampling problem,

$$\frac{\mathbf{E}\left[\mathcal{A}(\Delta V, N, q(\theta|\theta')/q(\theta'|\theta), c_1, \ldots, c_N, C, M(\theta, \theta'))\right]}{\mathbf{E}\left[\mathcal{A}(-\Delta V, N, q(\theta'|\theta)/q(\theta|\theta'), c_1, \ldots, c_N, C, M(\theta, \theta'))\right]} \neq \frac{\mu(\theta') \cdot q(\theta|\theta')}{\mu(\theta) \cdot q(\theta'|\theta)}. \tag{4}$$

Without loss of generality, assume that

$$q(\theta|\theta')/q(\theta'|\theta) \leq 1.$$

(This is without loss of generality since we can ensure it is the case by swapping $\theta$ and $\theta'$.) We fixed $\theta$ and $\theta'$ to be the pair satisfying Equation 4 throughout this section.

**Constructing an example.**  We use this to prove the theorem by a constructive example. Let $x_1, \ldots, x_N$ be defined by

$$x_i = \Delta V(i) = V_i(\theta) - V_i(\theta').$$

Define $X$ as the sum

$$X = \sum_{i=1}^{N} x_i.$$

For some parameter $K \in \mathbb{N}$ (to be defined later), consider the state space $\Omega$ defined as

$$\Omega = \{(k, z) \mid k \in \{0, \ldots, K-1\}, \ 0 \leq z \leq \exp(kX)\},$$

using the natural measure for a finite disjoint union of measure spaces. Define a target distribution over $\Omega$ given by the density

$$\pi(k, z) \propto \exp\left(-\sum_{i=1}^{N} k \cdot x_i\right),$$

or equivalently

$$\pi(k, z) \propto \exp\left(-\sum_{i=1}^{N} U_i(k, z)\right) \text{ where } U_i(k, z) = kx_i.$$

Define a proposal distribution $\hat{q}$, such that, starting from $(k, z)$:

- With probability $1/4$, we sample $z'$ uniformly from $[0, \exp(kX)]$ and propose a transition to $(k, z')$.

- With probability $1/4$, we propose a transition to $(k - 1, z)$, if it is in $\Omega$.

- With probability $\frac{1}{4} \cdot \frac{q(\theta|\theta')}{q(\theta'|\theta)}$, we propose a transition to $(k + 1, z)$, if it is in $\Omega$.

- With the remaining probability, we just propose to stay at $(k, z)$.

This is effectively acting as a random walk over $k$, and our goal will be to show that while the true target distribution $\pi$ has a marginal in $k$ that is the uniform distribution, the minibatch MH method causes the chain's transition to be biased to step more in one direction than another, resulting in a highly biased stationary distribution (where we can make the bias arbitrarily large by setting $K$).

We use the same $c_i$ and $C$ as before, and define a new function $\hat{M}$ such that

$$\hat{M}((k, z), (k + 1, z)) = \hat{M}((k, z), (k - 1, z)) = M(\theta, \theta')$$

and $\hat{M}(\cdots) = 0$ for other proposed transitions (we can set $\hat{M}$ however we want for pairs of states that are never proposed in a transition, since this will not affect the algorithm). Clearly, this setup satisfies Assumption 1, since the original distribution did.

Now, consider what our minibatch MH method will do when run on this task. There are three cases to consider.

**Proposed changes in $z$.** When a proposed change in $z$ is made, the resulting $\Delta U$ will be uniformly $0$, and the probability of the reverse transition will be equal ($1/4$ in both directions), so the algorithm will be passed the arguments

$$\mathcal{A}(0, N, 1, c_1, \ldots, c_N, C, 0).$$

Since this does not depend at all on $z$ or $k$, this means that the acceptance probability of these transitions will be the same regardless of the state. Call this probability $\alpha_0$.

**A proposal to decrease $k$.** When a proposal is made to decrease $k$, the probability of the forward and reverse transitions will be

$$\hat{q}((k - 1, z)|(k, z)) = \frac{1}{4} \text{ and } \hat{q}((k, z)|(k - 1, z)) = \frac{1}{4} \cdot \frac{q(\theta|\theta')}{q(\theta'|\theta)}.$$

It follows that

$$\frac{\hat{q}((k, z)|(k - 1, z))}{\hat{q}((k - 1, z)|(k, z))} = \frac{q(\theta|\theta')}{q(\theta'|\theta)}.$$

The energy function difference for this proposal will be

$$\Delta U(i) = U_i((k, z)) - U_i((k - 1, z)) = kx_i - (k - 1)x_i = x_i,$$

so in particular $\Delta U = \Delta V$. And, of course for this transition $\hat{M}$ will take on the value $M(\theta, \theta')$. So, the minibatch MH algorithm will be passed the arguments

$$\mathcal{A}(\Delta V, N, q(\theta|\theta')/q(\theta'|\theta), c_1, \ldots, c_N, C, M(\theta, \theta')),$$

and so it will accept with probability

$$\mathbf{E}\left[\mathcal{A}(\Delta V, N, q(\theta|\theta')/q(\theta'|\theta), c_1, \ldots, c_N, C, M(\theta, \theta'))\right].$$

Call this probability $\alpha_-$.

**A proposal to increase $k$.** When a proposal is made to increase $k$, the probability of the forward and reverse transitions will be

$$\hat{q}((k+1,z)|(k,z)) = \frac{1}{4} \cdot \frac{q(\theta|\theta')}{q(\theta'|\theta)}. \text{ and } \hat{q}((k,z)|(k+1,z)) = \frac{1}{4}.$$

It follows that

$$\frac{\hat{q}((k,z)|(k+1,z))}{\hat{q}((k+1,z)|(k,z))} = \frac{q(\theta'|\theta)}{q(\theta|\theta')}.$$

The energy function difference for this proposal will be

$$\Delta U(i) = U_i((k,z)) - U_i((k+1,z)) = kx_i - (k+1)x_i = -x_i,$$

so in particular $\Delta U = -\Delta V$. And, as before for this transition $\hat{M}$ will take on the value $M(\theta, \theta')$. So, the minibatch MH algorithm will be passed the arguments

$$\mathcal{A}(-\Delta V, N, q(\theta'|\theta)/q(\theta|\theta'), c_1, \ldots, c_N, C, M(\theta, \theta')),$$

and so it will accept with probability

$$\mathbf{E}\left[\mathcal{A}(-\Delta V, N, q(\theta'|\theta)/q(\theta|\theta'), c_1, \ldots, c_N, C, M(\theta, \theta'))\right].$$

Define the probability $\alpha_+$ as

$$\alpha_+ = \mathbf{E}\left[\mathcal{A}(-\Delta V, N, q(\theta'|\theta)/q(\theta|\theta'), c_1, \ldots, c_N, C, M(\theta, \theta'))\right] \cdot \frac{q(\theta|\theta')}{q(\theta'|\theta)}.$$

**The resulting Markov chain.** From the above analysis, we can conclude that the Markov chain that results from subsampling algorithm $\mathcal{A}$ applied to this method is as follows. Starting from $(k, z)$, if we let $\hat{T}$ denote the transition operator of this Markov chain,

- With probability $\frac{1}{4} \cdot \alpha_0$, we sample $z'$ uniformly from $[0, \exp(kX)]$ and transition to $(k, z')$.

- With probability $\frac{1}{4} \cdot \alpha_-$, we transition to $(k-1, z)$, if it is in $\Omega$.

- With probability $\frac{1}{4} \cdot \alpha_+$, we transition to $(k+1, z)$, if it is in $\Omega$.

- With the remaining probability, we just stay at $(k, z)$.

Consider the distribution

$$\nu(k, z) \propto \left(\frac{\alpha_+}{\alpha_-}\right)^k.$$

It is easy to see that this Markov chain satisfies detailed balance with $\nu$ as its stationary distribution. In particular,

$$\begin{aligned}
\nu(k, z) \cdot T((k-1,z)|(k,z)) &= \left(\frac{\alpha_+}{\alpha_-}\right)^k \cdot \frac{1}{4} \cdot \alpha_- \\
&= \left(\frac{\alpha_+}{\alpha_-}\right)^{k-1} \cdot \frac{1}{4} \cdot \alpha_+ \\
&= \nu(k-1, z) \cdot T((k,z)|(k-1,z)).
\end{aligned}$$

So $\nu$ will be a stationary distribution of the minibatch MH chain $\hat{T}$.

Observe that the marginal distribution of $k$ in $\pi$ is

$$\pi(k) = \int_0^{\exp(kX)} \pi(k, z) \, dz \propto \exp\left(-\sum_{i=1}^{N} k \cdot x_i\right) \cdot \exp(kX) = 1,$$

so the marginal distribution of $k$ in the target distribution is actually the uniform distribution. On the other hand, using the same derivation, the marginal distribution of $k$ in $\nu$ is

$$\nu(k) \propto \left(\frac{\alpha_+}{\alpha_-}\right)^k \cdot \exp(kX) = \left(\frac{\alpha_+}{\alpha_-} \cdot \exp(X)\right)^k.$$

We know immediately by substituting our definitions of $\alpha_+$ and $\alpha_-$ into (4) that

$$\frac{\alpha_-}{\alpha_+} \neq \frac{\mu(\theta')}{\mu(\theta)} = \exp\left(\sum_{i=1}^{N}(V_i(\theta) - V_i(\theta'))\right) = \exp\left(\sum_{i=1}^{N} x_i\right) = \exp(X).$$

As a consequence, we know that

$$\frac{\alpha_+}{\alpha_-} \cdot \exp(X) \neq 1.$$

Call this constant

$$A = \frac{\alpha_+}{\alpha_-} \cdot \exp(X),$$

and observe that $A \neq 1$ and that $A$ is independent of our choice of $K$ (which still remains unset). This gives

$$\nu(k) \propto A^k.$$

Explicitly, this distribution will be

$$\nu(k) = \frac{1}{\sum_{k=0}^{K-1} A^k} \cdot A^k = \frac{1-A}{1-A^K} \cdot A^k.$$

Since the total variation distance between two probability measures is lower bounded by the TV-distance between their marginal distributions in any one variable, and similarly the KL divergence is *also* lower bounded by the KL divergence between its marginal distributions in any one variable (both these facts follow directly from the monotonicity property of the $f$-divergence, of which the KL-divergence and TV-distance are both instances), to prove this theorem it suffices to show both TV-distance and KL-divergence bounds on the marginal distributions in $k$. We do this now.

**Bounding the total variation distance.** Now, we compute the total variation distance between $\pi$ and $\nu$. For this bit of the proof, we will just consider the marginal distribution in $k$, as this provides a lower bound on the TV distance between the joint distribution. For simplicity, for the rest of the proof, we let $\tilde{\pi}$ denote this marginal distribution of $k$ in $\nu$, and also let $\pi$ denote the marginal distribution of $k$ in $\pi$. By the definition of total variation distance,

$$\text{TV}(\pi, \tilde{\pi}) = \frac{1}{2} \sum_{k=0}^{K-1} |\tilde{\pi}(k) - \pi(k)|$$

$$= \frac{1}{2} \sum_{k=0}^{K-1} \left| \frac{1-A}{1-A^K} \cdot A^k - \frac{1}{K} \right|.$$

If $A < 1$,

$$\text{TV}(\pi, \tilde{\pi}) = \sum_{k=0}^{K_0} \left( \frac{1-A}{1-A^K} \cdot A^k - \frac{1}{K} \right)$$

$$= \frac{1-A^{K_0}}{1-A^K} - \frac{K_0}{K} \tag{5}$$

where $K_0$ is the largest $k$ such that

$$\frac{1-A}{1-A^K} \cdot A^k > \frac{1}{K}.$$

By solving the above equation, we have

$$K_0 = \left\lfloor \frac{\log(1-A^K) - \log(1-A) - \log(K)}{\log(A)} \right\rfloor.$$

We can lower bound $K_0$ by

$$K_0 \geq \frac{\log(1-A^K) - \log(1-A) - \log(K)}{\log(A)} - 1$$

$$\geq \frac{-\log(1-A) - \log(K)}{\log(A)} - 1.$$

It follows that the first term in (5) becomes

$$\frac{1 - A^{K_0}}{1 - A^K} \geq \frac{1 - \frac{1}{KA(1-A)}}{1 - A^K} \geq 1 - \frac{1}{KA(1-A)}.$$

We can also upper bound $K_0$ and then the second term can be bounded as the following

$$\frac{K_0}{K} \leq \frac{\log(1 - A^K) - \log(K)}{K \log(A)}.$$

When $K \geq \frac{\log\left(1 - \exp\left(-\frac{1}{2}\right)\right)}{\log(A)}$, we have $\log(1 - A^K) \geq -\frac{1}{2}$. Since $\log(K) \leq K^{\frac{1}{2}}$ and $K^{-1} \leq K^{-\frac{1}{2}}$, we have

$$\frac{K_0}{K} \leq \frac{-\frac{1}{2}K^{-1} - K^{-\frac{1}{2}}}{\log(A)} \leq -\left(\frac{3}{2\log(A)}\right) K^{-\frac{1}{2}}.$$

Therefore, the TV distance is bounded by

$$\mathrm{TV}(\pi, \tilde{\pi}) \geq 1 - \frac{1}{KA(1-A)} + \left(\frac{3}{2\log(A)}\right) K^{-\frac{1}{2}}$$

$$\geq 1 + \left(\frac{3}{2\log(A)} - \frac{1}{A(1-A)}\right) K^{-\frac{1}{2}}.$$

To make $\mathrm{TV}(\pi, \tilde{\pi}) \geq \delta$, we just need to set

$$K \geq \frac{\left(\frac{3}{2\log(A)} - \frac{1}{A(1-A)}\right)^2}{(1 - \delta)^2}.$$

Similarly, if $A > 1$,

$$\mathrm{TV}(\pi, \tilde{\pi}) = \sum_{k=K_0}^{K-1} \left(\frac{1 - A}{1 - A^K} \cdot A^k - \frac{1}{K}\right)$$

$$= \frac{A^K - A^{K_0}}{A^K - 1} - \frac{K - K_0}{K}$$

$$= \frac{K_0}{K} - \frac{A^{K_0} - 1}{A^K - 1}$$

where

$$K_0 = \left\lceil \frac{\log(A^K - 1) - \log(A - 1) - \log(K)}{\log(A)} \right\rceil$$

which is the smallest $k$ such that

$$\frac{1 - A}{1 - A^K} \cdot A^k > \frac{1}{K}.$$

We can get an upper bound of $K_0$ by

$$K_0 \leq \frac{\log(A^K - 1) - \log(A - 1) - \log(K)}{\log(A)} + 1$$

$$= \log_A\left(\frac{A^K - 1}{K(A - 1)}\right) + 1.$$

Therefore,

$$\frac{A^{K_0} - 1}{A^K - 1} \leq \frac{A \cdot \left(\frac{A^K - 1}{K(A-1)}\right) - 1}{A^K - 1}$$

$$= \frac{A}{K(A - 1)} - \frac{1}{A^K - 1}.$$

We can lower bound $K_0$ by

$$K_0 \geq \log_A\left(A^K - 1\right) - \log_A(A - 1) - \log_A(K).$$

When $K \geq 1 - \log_A(A - 1)$, $A^K - 1 \geq A^{K-1}$. Then we have

$$K_0 \geq \log_A\left(A^{K-1}\right) - \log_A(A - 1) - \log_A(K)$$
$$= K - 1 - \log_A(A - 1) - \log_A(K).$$

It follows that

$$\frac{K_0}{K} \geq 1 - \frac{1}{K} - \frac{\log_A(A - 1)}{K} - \frac{\log_A(K)}{K}.$$

Since $\log(K) \leq K^{\frac{1}{2}}$ and $K^{-1} \leq K^{-\frac{1}{2}}$, the TV distance can be bounded by

$$\mathrm{TV}(\pi, \tilde{\pi}) \geq 1 - \frac{1}{K} - \frac{\log_A(A - 1)}{K} - \frac{\log_A(K)}{K} - \frac{A}{K(A - 1)} + \frac{1}{A^K - 1}$$
$$\geq 1 - \left(1 + \log_A(A - 1) + \frac{1}{\log(A)} + \frac{A}{A - 1}\right) K^{-\frac{1}{2}}.$$

To make $\mathrm{TV}(\pi, \tilde{\pi}) \geq \delta$, we just need

$$K \geq \left(\frac{1 + \log_A(A - 1) + \frac{1}{\log(A)} + \frac{A}{A-1}}{1 - \delta}\right)^2.$$

Since we could set $K$ arbitrarily, it is clear that we can do this.

**Bounding the KL divergence.** We can compute KL divergence between $\pi$ and $\tilde{\pi}$ as follows

$$\mathrm{KL}(\pi, \tilde{\pi}) = \sum_{k=0}^{K-1} \frac{1}{K} \cdot \log\left(\frac{1}{K} \cdot \frac{1 - A^K}{(1 - A)A^k}\right)$$
$$= \frac{1}{K} \cdot \sum_{k=0}^{K-1} \left[\log\left(\frac{1}{K} \cdot \frac{1 - A^K}{(1 - A)}\right) - k\log(A)\right]$$
$$= \log\left(\frac{1 - A^K}{K(1 - A)}\right) - \frac{\log(A)}{K} \sum_{k=0}^{K-1} k$$
$$= \log\left(\frac{1 - A^K}{K(1 - A)}\right) - \frac{(K - 1)\log(A)}{2}$$

If $A < 1$, we have

$$\mathrm{KL}(\pi, \tilde{\pi}) = \log\left(1 - A^K\right) - \log((1 - A)K) - \frac{K\log(A)}{2} + \frac{\log(A)}{2}$$
$$\geq \log\left(1 - A^K\right) - \left(\frac{1 - A + \log(A)}{2}\right)K + \frac{\log(A)}{2}.$$

The last equation is because $\log(x) \leq \frac{x}{2}$.

To further simplify the above equation, we first note that $1 - A + \log(A) < 0$ when $A \neq 1$. And then when $K \geq \log_A\left(1 - A^{\frac{1}{2}}\right)$, we have $1 - A^K \geq A^{\frac{1}{2}}$. It follows that we can simplify it to be

$$\mathrm{KL}(\pi, \tilde{\pi}) \geq \log(A) - \left(\frac{1 - A + \log(A)}{2}\right)K.$$

To make $\mathrm{KL}(\pi, \tilde{\pi}) \geq \rho$, it is clear that we just need to set

$$K \geq \frac{2(\rho - \log(A))}{A - 1 - \log(A)}.$$

Consider when $A > 1$,

$$\text{KL}(\pi, \tilde{\pi}) = \log\left(\frac{A^K - 1}{K(A-1)}\right) - \frac{(K-1)\log(A)}{2}.$$

If $K \geq \frac{\log(2)}{\log(A)}$, we have that $A^K - 1 \geq \frac{A^K}{2}$. It follows that

$$\text{KL}(\pi, \tilde{\pi}) \geq K\log(A) - \log(K) - \log(2A - 2) - \frac{K\log(A)}{2}$$

$$= \frac{K\log(A)}{2} - \log(K) - \log(2A - 2).$$

To make $\text{KL}(\pi, \tilde{\pi}) \geq \rho$, we need

$$\frac{K\log(A)}{2} - \log(K) \geq \rho + \log(2A - 2).$$

Let $K = \exp(y)$. By Taylor series, we know $\exp(y) \geq \frac{y^2}{2}$. Then it follows that

$$\frac{y^2\log(A)}{4} - y \geq \rho + \log(2A - 2).$$

Solve the above inequality, we can get

$$y \geq \frac{1 + 2 \cdot \frac{\log(A)}{4} \cdot \left(\rho + \log(2A - 2)\right)}{2 \cdot \frac{\log(A)}{4}} = \frac{2 + \log(A)\left(\rho + \log(2A - 2)\right)}{\log(A)}.$$

It follows that it suffices to set

$$K \geq \exp\left(\frac{2 + \log(A)\left(\rho + \log(2A - 2)\right)}{\log(A)}\right).$$

**Concluding the proof.** The theorem now follows from choosing a $K$ large enough that both the TV distance inequality we derived and the KL divergence inequality we derived are satisfied. $\square$

## B Connection between Theorem 1 and TV Bound of Inexact MH Methods

Some inexact methods such as MHSubLhd [6] have bounded TV distance between the target distribution and the approximate distribution (see Proposition 3.2 in Bardenet et al. [6]). We would like to emphasize that Theorem 1 is compatible with these results. Specifically, Proposition 3.2 assumes $P_{\text{MH}}$ has a bounded mixing time. It is well known that this produces a TV bound for any kernel by coupling [18]. Our theorem does not have this assumption; it suggests that for MHSubLhd, with a given user-specified error, there exists a target distribution and proposal satisfying Theorem 1, on which $P_{\text{MH}}$ either does not have bounded mixing time or the mixing time is large enough such that the TV bound is greater than $\delta$.

## C Proof of Statement 1

*Proof.* We prove this by construction. Consider a dataset $\{x_i\}_{i=1}^N$. The data instances can take two values $\{-\frac{M}{N}, \frac{M}{N}\}$ where $M$ is a positive constant. Assume that half of the data instances take value $\frac{M}{N}$ and the remaining take $-\frac{M}{N}$. Let the target distribution be $\pi(\theta) = \frac{1}{Z}\exp\left(\theta \cdot \sum_{i=1}^N x_i\right)$ and the domain for $\theta$ be $\{0, 1, \ldots, K-1\}$. We define the proposal distribution to be the following

$$p(\theta, \theta) = \frac{1}{2}, \quad \text{for all } \theta; \quad p(\theta, \theta - 1) = \frac{1}{4}, \quad p(\theta, \theta + 1) = \frac{1}{4} \quad \text{for } \theta \in \{1, \ldots, K-2\};$$

and $p(0, 1) = p(K - 1, K - 2) = \frac{1}{2}$.

Recall that FMH factorizes the target distribution $\pi(\theta)$ and the proposal distribution $p(\theta)$ as follows

$$\pi(\theta) \propto \prod_{i=1}^{m} \pi_i(\theta), \qquad p(\theta) \propto \prod_{i=1}^{m} p_i(\theta)$$

where $m \geq 1$ and $\pi_i$ and $p_i$ are some non-negative functions. Then the acceptance rate is given by

$$a_{\text{FMH}}(\theta, \theta') = \prod_{i=1}^{m} \min\left(1, \frac{\pi(\theta')p_i(\theta', \theta)}{\pi(\theta)p_i(\theta, \theta')}\right).$$

A common choice is to set $m = N$. On this example, we can write the acceptance rate of transitioning from $\theta$ to $\theta' = \theta + 1$ in FMH as follows

$$a_{\text{FMH}}(\theta, \theta') = \prod_{i=1}^{N} \min\left(1, \exp(x_i)\right) = \left(\exp\left(-\frac{M}{N}\right)\right)^{\frac{N}{2}} = \exp\left(-\frac{M}{2}\right).$$

It is easy to show that the acceptance rate of transitioning from $\theta$ to $\theta' = \theta - 1$ in FMH is the same.

When $M > -2\log(p)$, it is clear that the acceptance rate of FMH is less than $p$. By contrast, the acceptance rate of standard MH is

$$a_{\text{MH}}(\theta, \theta') = \min\left(1, \exp\left(\pm \sum_{i=1}^{N} x_i\right)\right) = 1.$$

In order to preserve geometric ergodicity, Cornish et al. [12] introduces *truncated FMH* (TFMH) which forces FMH degrade to standard MH when the energy exceeds a threshold $R$. If we set hyperparameter $R > M/2$, then in each step, the value of $a_{\text{TFMH}}$ will be the same as $a_{\text{FMH}}$. Therefore, if setting $M > -2\log(p)$, we have

$$\frac{a_{\text{TFMH}}}{a_{\text{MH}}} \leq \frac{p}{1} = p.$$

If we set $R \leq M/2$, TFMH falls back to standard, full-batch MH — using the whole dataset at each step. This proves the statement. $\qquad\square$

## D  Construction of Algorithm 2

Algorithm 2 can be derived by carefully replacing the global bounds on the energy in PoissonMH [27] with local bounds on the energy differences (Assumption 1). PoissonMH is a variant of Poisson Gibbs and therefore inherits the same assumptions for Gibbs sampling on graphical models, which are often violated in the applications of MH. In particular, PoissonMH works on *factor graphs* which define a distribution $\pi(\theta)$ over a set of factors $\{\phi_i(\theta)\}_{i=1}^{N}$ as follows

$$\pi(\theta) \propto \exp\left(\sum_{i=1}^{N} \phi_i(\theta)\right).$$

PoissonMH assumes that each factor $\phi_i$ is non-negative without the loss of generality (we can add a positive constant to $\phi_i$ to make it non-negative without changing the distribution) and is bounded globally by a constant $M_i$. That is

$$0 \leq \phi_i(\theta) \leq M_i \text{ for all } \theta.$$

This assumption does not hold for most applications of MH, such as the linear and logistic regression experiments in Section 5.

Let $L = \sum_i M_i$ and define Poisson auxiliary variable $s_i$ as the following

$$s_i|\theta \sim \text{Poisson}\left(\frac{\lambda M_i}{L} + \phi_i(\theta)\right),$$

---

**Algorithm 3** PoissonMH

> **given:** initial state $\theta \in \Theta$; proposal dist. $q$; hyperparameter $\lambda$; Global bounds $M_i$, $L$
> **loop**
>     **propose** $\theta' \sim q(\cdot|\theta)$
>     **for** $i \in \{1, \ldots, N\}$ **do**
>         **sample** $s_i \sim \text{Poisson}\left(\frac{\lambda M_i}{L} + \phi_i(\theta)\right)$
>     **end for**
>     **form minibatch** $\mathcal{S} \leftarrow \{i | s_i > 0\}$
>     **compute MH ratio** $r \leftarrow \dfrac{\exp\left(\sum_{i \in \mathcal{S}} s_i \log\left(1 + \frac{L}{\lambda M_i} \phi_i(\theta')\right)\right) q(\theta'|\theta)}{\exp\left(\sum_{i \in \mathcal{S}} s_i \log\left(1 + \frac{L}{\lambda M_i} \phi_i(\theta)\right)\right) q(\theta|\theta')}$
>     **with probability** $\min(1, r)$, set $\theta \leftarrow \theta'$
> **end loop**

---

where $\lambda > 0$ is a hyperparameter. Running standard MH on the joint distribution of $\theta$ and $s_i$ results in the following acceptance ratio

$$r_{\text{PoissonMH}}(\theta, \theta') = \frac{\exp\left(\sum_i s_i \log\left(1 + \frac{L}{\lambda M_i} \phi_i(\theta')\right)\right) q(\theta'|\theta)}{\exp\left(\sum_i s_i \log\left(1 + \frac{L}{\lambda M_i} \phi_i(\theta)\right)\right) q(\theta|\theta')}.$$

Here, the sum is essentially performed over the set of index $i$ whose $s_i$ is greater than zero. When $s_i = 0$, it is clear that the factor $\phi_i$ will not appear in the acceptance ratio $r_{\text{PoissonMH}}$. Thus PoissonMH enables using a subset of factors for the MH decision step (Algorithm 3).

To construct our method from this, we can define the factor $\phi_i$ in the factor graph to be

$$\phi_i(x) = \frac{U_i(\theta) + U_i(\theta')}{2} - U_i(x) + \frac{c_i}{2} M(\theta, \theta') \tag{6}$$

where $x \in \{\theta, \theta'\}$. It is easy to see that $\phi_i$ satisfy $0 \leq \phi_i(x) \leq c_i M(\theta, \theta')$. And then we define the Poisson variables $s_i$ as the follows

$$s_i|(\theta, \theta') \sim \text{Poisson}\left(\frac{\lambda c_i}{C} + \phi_i(\theta)\right) = \text{Poisson}\left(\frac{\lambda c_i}{C} + \frac{U_i(\theta') - U_i(\theta) + c_i M(\theta, \theta')}{2}\right).$$

These Poisson auxiliary variables $\{s_i\}_{i=1}^N$ are called *local*, because their distributions change each iteration depending on the current pair $(\theta, \theta')$ and only rely on local bounds in Assumption 1. This is in contrast to the *global* auxiliary variables used in PoissonMH and FlyMC which are used to form a joint distribution with $\theta$ and both require global bounds in their conditional distributions.

The acceptance ratio $r_{\text{TunaMH}}$ is the same as $r_{\text{PoissonMH}}$ but with the new definitions of $s_i$ and $\phi_i$. We outline TunaMH using the notation of $\phi_i$ and $s_i$ in Algorithm 4.

We now show that Algorithm 4 is statistically equivalent to Algorithm 2. To see this, we first use *thinning*, a commonly used technique [7, 8, 12, 19, 27], to quickly resample all $s_i$ from their new distributions in each iteration in Algorithm 4. This is achieved by replacing the global bounds with the local bounds in Algorithm 4 in the Appendix of Zhang and De Sa [27]. Specifically, we first sample $B$ from a Poisson distribution

$$B \sim \text{Poisson}(\lambda + CM(\theta, \theta')).$$

Here $\lambda + CM(\theta, \theta')$ is an upper bound on $\mathbf{E}[\sum_i s_i]$. We then form the minibatch by running

> **for** $b \in \{1, \ldots, B\}$ **do**
>     **sample** $i_b$ such that $\mathbf{P}(i_b = i) = c_i/C$, for $i = 1 \ldots N$
>     **with probability** $\frac{\lambda c_{i_b} + C\phi_{i_b}(\theta)}{\lambda c_{i_b} + Cc_{i_b} M(\theta, \theta')}$ **add** $i_b$ to $\mathcal{I}$
> **end for**

By substituting $\lambda = \chi C^2 M^2(\theta, \theta')$ and the expression of $\phi_i$, we can get the part of "form minibatch $\mathcal{I}$" in Algorithm 2.

---
**Algorithm 4** TunaMH
---

**given:** initial state $\theta \in \Theta$; proposal dist. $q$; $\lambda$; Asm. 1 parameters $c_i$, $C$, $M$; function $\phi_i$ defined in (6)
**loop**
    **propose** $\theta' \sim q(\cdot|\theta)$ and **compute** $M(\theta, \theta')$
    **for** $i \in \{1, \dots, N\}$ **do**
        **sample** $s_i \sim \text{Poisson}\left(\frac{\lambda c_i}{C} + \phi_i(\theta)\right)$
    **end for**
    **form minibatch** $\mathcal{S} \leftarrow \{i|s_i > 0\}$
    **compute MH ratio** $r \leftarrow \dfrac{\exp\left(\sum_{i \in \mathcal{S}} s_i \log\left(1 + \frac{C}{\lambda c_i}\phi_i(\theta')\right)\right) q(\theta'|\theta)}{\exp\left(\sum_{i \in \mathcal{S}} s_i \log\left(1 + \frac{C}{\lambda c_i}\phi_i(\theta)\right)\right) q(\theta|\theta')}$
    **with probability** $\min(1, r)$, set $\theta \leftarrow \theta'$
**end loop**

---

To see that the MH ratio in Algorithm 2 and 4 are equivalent, we can write out $r$ in Algorithm 4 using the above fast way of resampling $s_i$

$$r_{\text{TunaMH}} = \frac{\exp\left(\sum_{i \in \mathcal{I}} \log\left(1 + \frac{C}{\lambda c_i}\phi_i(\theta')\right)\right) q(\theta'|\theta)}{\exp\left(\sum_{i \in \mathcal{I}} \log\left(1 + \frac{C}{\lambda c_i}\phi_i(\theta)\right)\right) q(\theta|\theta')}.$$

We then substitute the definition of $\phi_i$ in (6) and it follows that

$$r_{\text{TunaMH}} = \exp\left(\sum_{i \in \mathcal{I}}\left(\log\left(\frac{2\lambda c_i + C\left(U_i(\theta) - U_i(\theta') + c_i M(\theta, \theta')\right)}{2\lambda c_i + C\left(U_i(\theta') - U_i(\theta) + c_i M(\theta, \theta')\right)}\right)\right)\right) \cdot \frac{q(\theta'|\theta)}{q(\theta|\theta')}.$$

We can rearrange the $\log$ term inside $r_{\text{TunaMH}}$ as

$$\log\left(\frac{2\lambda c_i + C\left(U_i(\theta) - U_i(\theta') + c_i M(\theta, \theta')\right)}{2\lambda c_i + C\left(U_i(\theta') - U_i(\theta) + c_i M(\theta, \theta')\right)}\right)$$
$$= \log\left(\frac{2\lambda c_i + C\left(U_i(\theta) - U_i(\theta')\right) + c_i CM(\theta, \theta')}{2\lambda c_i + C\left(U_i(\theta') - U_i(\theta)\right) + c_i CM(\theta, \theta')}\right)$$
$$= \log\left(\frac{1 + \frac{C}{2\lambda c_i + c_i CM(\theta, \theta')}\left(U_i(\theta) - U_i(\theta')\right)}{1 + \frac{C}{2\lambda c_i + c_i CM(\theta, \theta')}\left(U_i(\theta') - U_i(\theta)\right)}\right)$$
$$= 2\operatorname{artanh}\left(\frac{C\left(U_i(\theta) - U_i(\theta')\right)}{c_i(2\lambda + CM(\theta, \theta'))}\right).$$

So $r_{\text{TunaMH}}$ can be written as

$$r_{\text{TunaMH}} = \exp\left(2\sum_{i \in \mathcal{I}}\operatorname{artanh}\left(\frac{C\left(U_i(\theta) - U_i(\theta')\right)}{c_i(2\lambda + CM(\theta, \theta'))}\right)\right) \cdot \frac{q(\theta'|\theta)}{q(\theta|\theta')}.$$

Finally setting $\lambda$ to be $\chi C^2 M^2(\theta, \theta')$ produces the MH ratio in Algorithm 2.

By proving the equivalence of the minibatch and the MH ratio, we show that Algorithm 2 and 4 are statistically equivalent.

# E  Proof of Theorem 2

In this section, we prove Theorem 2, which asserts that TunaMH is reversible and has stationary distribution $\pi$, and gives bounds on its spectral gap relative to the spectral gap of the original Metropolis-Hastings algorithm.

*Proof.* For convenience, we prove Theorem 2 using Algorithm 4 statement which is statistically equivalent to Algorithm 2. The transition operator can be written as the following

$$T(\theta, \theta')$$

$$= \mathbf{E}\left\{ q(\theta'|\theta) \min\left(1, \frac{q(\theta|\theta') \exp\left(\sum_i \left[s_i \log\left(\frac{\lambda c_i}{C} + \phi_i(\theta')\right) - \log s_i!\right]\right)}{q(\theta'|\theta) \exp\left(\sum_i \left[s_i \log\left(\frac{\lambda c_i}{C} + \phi_i(\theta)\right) - \log s_i!\right]\right)}\right)\right\}$$

$$= \mathbf{E}\left\{ q(\theta'|\theta) \min\left(1, \frac{q(\theta|\theta') \exp\left(\sum_i \left[s_i \log\left(\frac{\lambda c_i}{C} + \phi_i(\theta')\right)\right]\right)}{q(\theta'|\theta) \exp\left(\sum_i \left[s_i \log\left(\frac{\lambda c_i}{C} + \phi_i(\theta)\right)\right]\right)}\right)\right\}$$

$$= \sum_s \left\{ q(\theta'|\theta) \min\left(1, \frac{q(\theta|\theta') \exp\left(\sum_i \left[s_i \log\left(\frac{\lambda c_i}{C} + \phi_i(\theta')\right)\right]\right)}{q(\theta'|\theta) \exp\left(\sum_i \left[s_i \log\left(\frac{\lambda c_i}{C} + \phi_i(\theta)\right)\right]\right)}\right)\right\} \prod_i p(s_i|\theta, \theta')$$

$$= \sum_s \left\{ q(\theta'|\theta) \min\left( \exp\left(\sum_i \left[s_i \log\left(\frac{\lambda c_i}{C} + \phi_i(\theta)\right) - \phi_i(\theta) - \frac{\lambda c_i}{C} - \log s_i!\right]\right), \right.\right.$$

$$\left.\left. \frac{q(\theta|\theta') \exp\left(\sum_i \left[s_i \log\left(\frac{\lambda c_i}{C} + \phi_i(\theta')\right)\right]\right)}{q(\theta'|\theta) \exp\left(\sum_i \phi_i(\theta) + \frac{\lambda c_i}{C} + \log s_i!\right)}\right)\right\}$$

$$= \sum_s \left\{ q(\theta'|\theta) \min\left( \exp\left(\sum_i \left[s_i \log\left(\frac{\lambda c_i}{C} + \phi_i(\theta)\right) - \phi_i(\theta) - \frac{\lambda c_i}{C} - \log s_i!\right]\right), \right.\right.$$

$$\left.\left. \frac{q(\theta|\theta')}{q(\theta'|\theta)} \exp\left(\sum_i \left[s_i \log\left(\frac{\lambda c_i}{C} + \phi_i(\theta')\right) - \phi_i(\theta) - \frac{\lambda c_i}{C} - \log s_i!\right]\right)\right)\right\}$$

Multiplying $\pi(\theta)$ to both sides produces

$$\pi(\theta)T(\theta, \theta')$$

$$= \frac{1}{Z} \exp\left(-\sum_i U_i(\theta)\right) T(\theta, \theta')$$

$$= \frac{1}{Z} \sum_s \min\left( q(\theta'|\theta) \exp\left(\sum_i \left[s_i \log\left(\frac{\lambda c_i}{C} + \phi_i(\theta)\right)\right.\right.\right.$$

$$\left.\left. - \frac{U_i(\theta) + U_i(\theta')}{2} - \frac{c_i}{2}M(\theta, \theta') - \frac{\lambda c_i}{C} - \log s_i!\right]\right),$$

$$q(\theta|\theta') \exp\left(\sum_i \left[s_i \log\left(\frac{\lambda c_i}{C} + \phi_i(\theta')\right)\right.\right.$$

$$\left.\left.\left. - \frac{U_i(\theta) + U_i(\theta')}{2} - \frac{c_i}{2}M(\theta, \theta') - \frac{\lambda c_i}{C} - \log s_i!\right]\right)\right).$$

It is clear that the expression is symmetric in $\theta$ and $\theta'$. Therefore the chain is reversible and its stationary distribution is $\pi(\theta)$. This proves the first part of the theorem.

To prove the second part of the theorem, the bound on the spectral gap, we continue to reduce the transition probability in the previous proof to

$$\pi(\theta)T(\theta, \theta')$$

$$= \frac{1}{Z}\sum_s \min\left(q(\theta'|\theta)\exp\left(\sum_i \left[s_i \log\left(\frac{\lambda c_i}{C} + \phi_i(\theta)\right)\right.\right.\right.$$

$$\left.\left.- \frac{U_i(\theta) + U_i(\theta')}{2} - \frac{c_i}{2}M(\theta, \theta') - s_i \log\frac{\lambda c_i}{C}\right]\right),$$

$$q(\theta|\theta')\exp\left(\sum_i\left[s_i\log\left(\frac{\lambda c_i}{C} + \phi_i(\theta')\right)\right.\right.$$

$$\left.\left.\left.- \frac{U_i(\theta) + U_i(\theta')}{2} - \frac{c_i}{2}M(\theta, \theta') - s_i\log\frac{\lambda c_i}{C}\right]\right)\right)$$

$$\cdot \prod_i \frac{1}{s_i!}\exp\left(-\frac{\lambda c_i}{C}\right)\left(\frac{\lambda c_i}{C}\right)^{s_i}$$

$$= \frac{1}{Z}\sum_s\min\left(q(\theta'|\theta)\exp\left(\sum_i\left[s_i\log\left(1 + \frac{C}{\lambda c_i}\phi_i(\theta)\right)\right.\right.\right.$$

$$\left.\left.- \frac{U_i(\theta) + U_i(\theta')}{2} - \frac{c_i}{2}M(\theta, \theta')\right]\right),$$

$$q(\theta|\theta')\exp\left(\sum_i\left[s_i\log\left(1 + \frac{C}{\lambda c_i}\phi_i(\theta')\right) - \frac{U_i(\theta) + U_i(\theta')}{2} - \frac{c_i}{2}M(\theta, \theta')\right]\right)\right)$$

$$\cdot\prod_i\frac{1}{s_i!}\exp\left(-\frac{\lambda c_i}{C}\right)\left(\frac{\lambda c_i}{C}\right)^{s_i}.$$

Note that $s_i$ here are non-negative integers that a Poisson variable can take, not variables. So if we let $r_i \sim \text{Poisson}\left(\frac{\lambda c_i}{C}\right)$ and $r_i$ to be all independent, we can write this as

$$\pi(\theta)T(\theta, \theta') = \frac{1}{Z}\mathbf{E}\min\left(q(\theta'|\theta)\exp\left(\sum_i r_i\log\left(1 + \frac{C}{\lambda c_i}\phi_i(\theta)\right)\right),\right.$$

$$\left.q(\theta|\theta')\exp\left(\sum_i r_i\log\left(1 + \frac{C}{\lambda c_i}\phi_i(\theta')\right)\right)\right)$$

$$\cdot\exp\left[-\frac{1}{2}\left(\sum_i U_i(\theta) + \sum_i U_i(\theta') + CM(\theta, \theta')\right)\right].$$

Assume $G(\theta, \theta')$ is the transition operator of standard MH. Consider the ratio

$$\frac{\pi(\theta)T(\theta, \theta')}{\pi(\theta)G(\theta, \theta')}$$

$$= \frac{1}{Z}\mathbf{E}\min\left(q(\theta'|\theta)\exp\left(\sum_i r_i\log\left(1 + \frac{C}{\lambda c_i}\phi_i(\theta)\right)\right),\right.$$

$$\left.q(\theta|\theta')\exp\left(\sum_i r_i\log\left(1 + \frac{C}{\lambda c_i}\phi_i(\theta')\right)\right)\right)$$

$$\cdot\exp\left[-\frac{1}{2}\left(\sum_i U_i(\theta) + \sum_i U_i(\theta') + CM(\theta, \theta')\right)\right]$$

$$\cdot\left[1\left/\left(\frac{1}{Z}\min\left(q(\theta'|\theta)\exp\left(-\sum_i U_i(\theta)\right), q(\theta|\theta')\exp\left(-\sum_i U_i(\theta')\right)\right)\right)\right]\right..$$

We know that $\frac{\min(A,B)}{\min(C,D)} = \min\left(\frac{A}{\min(C,D)}, \frac{B}{\min(C,D)}\right) \geq \min\left(\frac{A}{C}, \frac{B}{D}\right)$. The last inequality is due to the fact that $\frac{1}{\min(C,D)} \geq \frac{1}{C}$ and $\frac{1}{\min(C,D)} \geq \frac{1}{D}$.

With this inequality, we can continue simplifying the ratio,

$$
\frac{\pi(\theta)T(\theta,\theta')}{\pi(\theta)G(\theta,\theta')}
$$

$$
\geq \mathbf{E}\left[\min\left(\frac{\exp\left(\sum_i r_i \log\left(1 + \frac{C}{\lambda c_i}\phi_i(\theta)\right)\right)}{\exp\left(-\sum_i U_i(\theta)\right)}, \frac{\exp\left(\sum_i r_i \log\left(1 + \frac{C}{\lambda c_i}\phi_i(\theta')\right)\right)}{\exp\left(-\sum_i U_i(\theta')\right)}\right)\right]
$$

$$
\cdot \exp\left[-\frac{1}{2}\left(\sum_i U_i(\theta) + \sum_i U_i(\theta') + CM(\theta,\theta')\right)\right]
$$

$$
= \mathbf{E}\left[\min\left(\exp\left(\sum_i \left(r_i \log\left(1 + \frac{C}{\lambda c_i}\phi_i(\theta)\right) - \phi_i(\theta)\right)\right),\right.\right.
$$

$$
\left.\left.\exp\left(\sum_i \left(r_i \log\left(1 + \frac{C}{\lambda c_i}\phi_i(\theta')\right) - \phi_i(\theta')\right)\right)\right)\right]
$$

$$
= \mathbf{E}\left[\max\left(\exp\left(\sum_i \left(\phi_i(\theta) - r_i \log\left(1 + \frac{C}{\lambda c_i}\phi_i(\theta)\right)\right)\right),\right.\right.
$$

$$
\left.\left.\exp\left(\sum_i \left(\phi_i(\theta') - r_i \log\left(1 + \frac{C}{\lambda c_i}\phi_i(\theta')\right)\right)\right)\right)^{-1}\right].
$$

Because $f(x) = \frac{1}{x}$ is a convex function, by Jensen's inequality it follows

$$
\frac{\pi(\theta)T(\theta,\theta')}{\pi(\theta)G(\theta,\theta')} \geq \mathbf{E}\left[\max\left(\exp\left(\sum_i \left(\phi_i(\theta) - r_i \log\left(1 + \frac{C}{\lambda c_i}\phi_i(\theta)\right)\right)\right),\right.\right.
$$

$$
\left.\left.\exp\left(\sum_i \left(\phi_i(\theta') - r_i \log\left(1 + \frac{C}{\lambda c_i}\phi_i(\theta')\right)\right)\right)\right)\right]^{-1}.
$$

We use $\max(A,B) \leq (A^p + B^p)^{\frac{1}{p}}$ to remove the $\max$ function.

$$
\frac{\pi(\theta)T(\theta,\theta')}{\pi(\theta)G(\theta,\theta')} \geq \mathbf{E}\left[\left(\exp\left(p\sum_i \left(\phi_i(\theta) - r_i \log\left(1 + \frac{C}{\lambda c_i}\phi_i(\theta)\right)\right)\right)+\right.\right.
$$

$$
\left.\left.\exp\left(p\sum_i \left(\phi_i(\theta') - r_i \log\left(1 + \frac{C}{\lambda c_i}\phi_i(\theta')\right)\right)\right)\right)^{\frac{1}{p}}\right]^{-1}.
$$

Since $x^{\frac{1}{p}}$ is concave, by Jensen's inequality

$$
\frac{\pi(\theta)T(\theta,\theta')}{\pi(\theta)G(\theta,\theta')} \geq \mathbf{E}\left[\exp\left(p\sum_i \left(\phi_i(\theta) - r_i \log\left(1 + \frac{C}{\lambda c_i}\phi_i(\theta)\right)\right)\right)+\right.
$$

$$
\left.\exp\left(p\sum_i \left(\phi_i(\theta') - r_i \log\left(1 + \frac{C}{\lambda c_i}\phi_i(\theta')\right)\right)\right)\right]^{-\frac{1}{p}}
$$

$$
= \left[\prod_i \mathbf{E}\exp\left(p\phi_i(\theta) - pr_i \log\left(1 + \frac{C}{\lambda c_i}\phi_i(\theta)\right)\right)+\right.
$$

$$
\left.\prod_i \mathbf{E}\exp\left(p\phi_i(\theta') - pr_i \log\left(1 + \frac{C}{\lambda c_i}\phi_i(\theta')\right)\right)\right]^{-\frac{1}{p}}.
$$

$\mathbf{E}\left[\exp\left(-pr_i\log\left(1+\frac{C}{\lambda c_i}\phi_i(\theta)\right)\right)\right]$ is the moment generating function of the Poisson random variable $r_i$ evaluated at

$$t = -p\log\left(1+\frac{C}{\lambda c_i}\phi_i(\theta)\right).$$

We know that

$$\mathbf{E}\exp(r_i t) = \exp\left(\frac{\lambda c_i}{C}\left(\exp(t)-1\right)\right),$$

therefore,

$$\mathbf{E}\left[\exp\left(-pr_i\log\left(1+\frac{C}{\lambda c_i}\phi_i(\theta)\right)\right)\right] = \exp\left(\frac{\lambda c_i}{C}\left(1+\frac{C}{\lambda c_i}\phi_i(\theta)\right)^{-p} - \frac{\lambda c_i}{C}\right).$$

Substituting this into the original expression produces

$$\frac{\pi(\theta)T(\theta,\theta')}{\pi(\theta)G(\theta,\theta')} \geq \left[\prod_i \exp\left(\frac{\lambda c_i}{C}\left(1+\frac{C}{\lambda c_i}\phi_i(\theta)\right)^{-p} - \frac{\lambda c_i}{C} + p\phi_i(\theta)\right)\right.$$

$$\left. + \prod_i \exp\left(\frac{\lambda c_i}{C}\left(1+\frac{C}{\lambda c_i}\phi_i(\theta')\right)^{-p} - \frac{\lambda c_i}{C} + p\phi_i(\theta')\right)\right]^{-\frac{1}{p}}.$$

Considering the term inside $\exp$. Define a function $f(y) = \frac{\lambda c_i}{C}\left(1+\frac{C}{\lambda c_i}y\right)^{-p} - \frac{\lambda c_i}{C} + py$ for $y \geq 0$. It is clear that $f(0) = 0$. The first derivative is

$$f'(y) = p + (-p)\left(1+\frac{C}{\lambda c_i}y\right)^{-p-1}$$

which is also 0 at $y = 0$. The second and third derivatives are

$$f''(y) = (-p)(-p-1)\frac{C}{\lambda c_i}\left(1+\frac{C}{\lambda c_i}y\right)^{-p-2}, \tag{7}$$

$$f'''(y) = (-p)(-p-1)(-p-2)\left(\frac{C}{\lambda c_i}\right)^2\left(1+\frac{C}{\lambda c_i}y\right)^{-p-3}. \tag{8}$$

By Taylor series, we have

$$f(y) = f(0) + f'(0)y + \frac{f''(0)}{2!}y^2 + \frac{f'''(v)}{3!}y^3$$

where $v$ is between 0 and $y$. By (8), we know that $f'''(v) \leq 0$, therefore since $y \geq 0$, we have

$$f(y) \leq f(0) + f'(0)y + \frac{f''(0)}{2!}y^2$$

$$= \frac{f''(0)}{2!}y^2.$$

Substituting $y = \phi_i(\theta)$ produces

$$f(\phi_i(\theta)) \leq (-p)(-p-1)\frac{C}{\lambda c_i}\phi_i^2(\theta)$$

$$\leq (-p)(-p-1)\frac{C}{\lambda c_i}c_i^2 M^2(\theta,\theta').$$

Similarly, we can get

$$f(\phi_i(\theta')) \leq p(p+1)\frac{C}{\lambda c_i}c_i^2 M^2(\theta,\theta').$$

Substituting these to the spectral ratio, we get

$$\frac{\pi(\theta)T(\theta,\theta')}{\pi(\theta)G(\theta,\theta')} \geq \left[2\prod_i \exp\left(p(p+1)\frac{C}{\lambda c_i}c_i^2 M^2(\theta,\theta')\right)\right]^{-\frac{1}{p}}$$

$$= \left[2\exp\left(\sum_i p(p+1)\frac{C}{\lambda}c_i M^2(\theta,\theta')\right)\right]^{-\frac{1}{p}}$$

$$= \left[2\exp\left(p(p+1)\frac{C^2}{\lambda}M^2(\theta,\theta')\right)\right]^{-\frac{1}{p}}$$

$$= 2^{-\frac{1}{p}}\exp\left(-(p+1)\frac{C^2}{\lambda}M^2(\theta,\theta')\right).$$

Now, we maximize the R.H.S. with respect to $p$. Let $E = \frac{C^2}{\lambda}M^2(\theta,\theta')$, then it becomes

$$2^{-\frac{1}{p}}\exp\left(-(p+1)E\right) = \exp\left(-E - pE - \frac{1}{p}\log 2\right).$$

The maximum is attained at $p = \sqrt{\frac{\log 2}{E}}$ and the value is

$$\exp\left(-E - 2\sqrt{E\log 2}\right).$$

It follows that

$$\frac{\pi(\theta)T(\theta,\theta')}{\pi(\theta)G(\theta,\theta')} \geq \exp\left(-\frac{C^2}{\lambda}M^2(\theta,\theta') - 2\sqrt{\frac{C^2}{\lambda}M^2(\theta,\theta')\log 2}\right).$$

We set $\lambda = \chi C^2 M^2(\theta,\theta')$, it becomes

$$\frac{\pi(\theta)T(\theta,\theta')}{\pi(\theta)G(\theta,\theta')} \geq \exp\left(-\frac{1}{\chi} - 2\sqrt{\frac{\log 2}{\chi}}\right).$$

We complete the theorem by a Dirichlet form argument. We can write the Dirichlet form $\mathcal{E}(f)$ of a Markov chain with transition operator $G$ as [13]:

$$\mathcal{E}(f) = \frac{1}{2}\int\int\left[(f(\theta) - f(\theta'))^2\right]G(\theta,\theta')\pi(\theta)d\theta d\theta'.$$

If we let $L_0^2(\pi)$ to be the Hilbert space of functions $f$ such that $f$ has mean zero and is square integrable with respect to probability measure $\pi$. It follows that the spectral gap $\gamma$ of a Markov chain is [2]

$$\gamma = \inf_{f\in L_0^2(\pi):Var_\pi[f]=1}\mathcal{E}(f).$$

From this, it is easy to get that

$$\bar{\gamma} = \inf_{f\in L_0^2(\pi):Var_\pi[f]=1}\left[\frac{1}{2}\int\int\left[(f(\theta) - f(\theta'))^2\right]T(\theta,\theta')\pi(\theta)d\theta d\theta'\right]$$

$$\geq \exp\left(-\frac{1}{\chi} - 2\sqrt{\frac{\log 2}{\chi}}\right)\cdot\inf_{f\in L_0^2(\pi):Var_\pi[f]=1}\left[\frac{1}{2}\int\int\left[(f(\theta) - f(\theta'))^2\right]G(\theta,\theta')\pi(\theta)d\theta d\theta'\right]$$

$$= \exp\left(-\frac{1}{\chi} - 2\sqrt{\frac{\log 2}{\chi}}\right)\cdot\gamma.$$

$\square$

# F  Derivation of Equation (2)

Based on the bound in Theorem 2, to make sure that the spectral ratio $\bar{\gamma}/\gamma \geq \kappa$, we can set $\chi$ such that

$$\exp\left(-\frac{1}{\chi} - 2\sqrt{\frac{\log 2}{\chi}}\right) = \kappa.$$

Solving the above equation gives us

$$\chi = \frac{(2\log 2 - \log \kappa + 2\sqrt{\log 2(\log 2 - \log \kappa)})}{\log^2 \kappa} \leq \frac{4}{(1-\kappa)\log(1/\kappa)}.$$

Since the spectral gap ratio is monotonically increasing w.r.t. $\chi$, we can instead set $\chi$ to the upper bound

$$\chi = \frac{4}{(1-\kappa)\log(1/\kappa)}$$

which guarantees that $\bar{\gamma}/\gamma \geq \kappa$.

# G  Theoretically Optimal Value of $\chi$

The overall wall-clock time $L$ for a chain to converge can be represented as the number of steps times the wall-clock time $l$ of each step. We then minimize an upper bound of this overall wall-clock time to get the optimal value of $\chi$.

Consider a lazy Markov chain on a finite state $\Theta$. The *relaxation time* $t_{\text{rel}}$ of a Markov chain is defined to be the inverse of the spectral gap $\gamma$: $t_{\text{rel}} = 1/\gamma$. The *mixing time* $t_{\text{mix}}$, i.e. the number of steps required for a chain to converge to within TV distance $\delta$ to the target distribution $\pi$, is bounded by Levin and Peres [18]

$$t_{\text{mix}} \leq t_{\text{rel}} \log\left(\frac{1}{\delta \cdot \min_{\theta \in \Theta} \pi(\theta)}\right).$$

It follows that the overall wall-clock time $L$ is upper bouned by

$$L = l \cdot t_{\text{mix}} \leq l \cdot t_{\text{rel}} \log\left(\frac{1}{\delta \cdot \min_{\theta \in \Theta} \pi(\theta)}\right).$$

We assume that the expected wall clock time to run a step is proportional to the batch size plus some constant, which measures the cost of computing the proposal. Specifically, We use $\eta$ and $\xi$ to denote the time to get a proposal $\theta'$ and compute a $U_i$ in a step. Then we can write the time of a step $l$ as

$$l = B\xi + \eta.$$

In order to minimize $L$, we can instead minimize its upper bound, which is equivalent to minimize

$$l \cdot t_{\text{rel}} = (B\xi + \eta) \cdot \frac{1}{\gamma}. \tag{9}$$

Recall that for TunaMH, the average batch size over all steps is

$$\mathbf{E}_{(\theta,\theta')\sim\pi(\theta)q(\theta'|\theta)}[\chi C^2 M^2(\theta,\theta') + CM(\theta,\theta')],$$

and the spectral gap $\bar{\gamma}$ is lower bounded by the spectral gap of standar MH $\gamma$ such that

$$\bar{\gamma} \geq \exp\left(-\frac{1}{\chi} - 2\sqrt{\frac{\log 2}{\chi}}\right) \cdot \gamma.$$

Substituting the expression of batch size and spectral gap to (9) gives

$$l \cdot t_{\text{rel}} \leq \left(\mathbf{E}_{(\theta,\theta')\sim\pi(\theta)q(\theta'|\theta)}[\chi C^2 M^2(\theta,\theta') + CM(\theta,\theta')]\xi + \eta\right) \cdot \exp\left(\frac{1}{\chi} + 2\sqrt{\frac{\log 2}{\chi}}\right) \cdot \frac{1}{\gamma}.$$

To minimize the RHS of the above equation over $\chi$, we let the derivative w.r.t. $\chi$ to be zero and get,

$$
\begin{aligned}
\xi C^2 \mathbf{E}_{(\theta,\theta')\sim\pi(\theta)q(\theta'|\theta)}[M^2(\theta,\theta')]\chi^{-1} &+ (\xi C \mathbf{E}_{(\theta,\theta')\sim\pi(\theta)q(\theta'|\theta)}[M(\theta,\theta')] + \eta)\chi^{-2} \\
&+ \sqrt{\log 2}\,\xi C^2 \mathbf{E}_{(\theta,\theta')\sim\pi(\theta)q(\theta'|\theta)}[M^2(\theta,\theta')]\chi^{-\frac{1}{2}} \\
&+ \sqrt{\log 2}(\xi C \mathbf{E}_{(\theta,\theta')\sim\pi(\theta)q(\theta'|\theta)}[M(\theta,\theta')] + \eta)\chi^{-\frac{3}{2}} \\
&= \xi C^2 \mathbf{E}_{(\theta,\theta')\sim\pi(\theta)q(\theta'|\theta)}[M^2(\theta,\theta')].
\end{aligned}
$$

When $\chi$ is small, the LHS is approximately $(\xi C \mathbf{E}_{(\theta,\theta')\sim\pi(\theta)q(\theta'|\theta)}[M(\theta,\theta')] + \eta)\chi^{-2}$ which gives us

$$
\chi = \sqrt{\frac{\xi C \mathbf{E}_{(\theta,\theta')\sim\pi(\theta)q(\theta'|\theta)}[M(\theta,\theta')] + \eta}{\xi C^2 \mathbf{E}_{(\theta,\theta')\sim\pi(\theta)q(\theta'|\theta)}[M^2(\theta,\theta')]}}.
$$

When it is quick to get a proposal ($\eta \approx 0$) and the variance of $M$ is small, we can further simplify it to

$$
\chi = \frac{1}{\sqrt{C \mathbf{E}_{(\theta,\theta')\sim\pi(\theta)q(\theta'|\theta)}[M(\theta,\theta')]}}.
$$

In practice, we can get the above theoretically optimal value of $\chi$ by empirically estimating the mean and variance of $M(\theta,\theta')$. Note that even if these empirical estimates are accurate, there may exist better $\chi$, since the upper bounds (the mixing time bound and the spectral gap bound) we use to get the optimal value may be loose. We give a simpler heuristic to tune $\chi$ in practice in Section 5.

## H   Proof of Theorem 3

First, we will show the following lemma, which gives half of what we want to have in the theorem.

**Lemma 1.** *Considering the same setting as the theorem, the average batch size $B$ of any exact, stateless minibatch MH algorithm at any iteration follows*

$$
\mathbf{E}[B] \geq 2^{-18} \cdot \kappa C^2 M^2(\theta,\theta') - 2^{-4} \cdot \kappa.
$$

*Proof.* We prove the lemma by construction. First, observe that since the state space $\Theta$ has at least two states, we can restrict our attention to just two of those states, by choosing a $\pi$ that has zero mass on any other state in the space and a $q$ that never proposes transitioning out to any of those other states (at which $\pi$ has zero mass). Such a proposal will still be ergodic, so it still satisfies our general assumption that we consider only ergodic chains in this paper. Without loss of generality, suppose that those two states are $\{-\frac{M}{2}, \frac{M}{2}\}$ (this is without loss of generality because we can always just rename the states), and let $C$ denote the constant in the theorem statement and define (with a bit of abuse of notation) the constant $M := M(-\frac{M}{2}, \frac{M}{2})$. By doing this, we can (again without loss of generality) restrict our attention to the case where $\Theta = \{-\frac{M}{2}, \frac{M}{2}\}$.

Next, we construct our counterexample. Let the dataset be $\{x_i\}_{i=1}^N$ where $x_i \in \{-1, 1\}$. We let the domain for parameter $\theta$ to be $\{-\frac{M}{2}, \frac{M}{2}\}$, and the target distribution to be

$$
\pi(\theta) = \frac{1}{Z} \exp\left(-\sum_{i=1}^N U_i(\theta)\right) = \frac{1}{Z} \exp\left(-\frac{C\theta}{N}\sum_{i=1}^N x_i\right)
$$

where $U_i(\theta) = \frac{C}{N} \cdot \theta x_i$. Note that by letting $N$ become large, any minibatch MH algorithm that queries the energy difference oracle some number of times will observe a distribution of energy differences that is arbitrarily close to a sequence of independent identically distributed random variables supported on $\{\pm\frac{CM}{N}\}$.

We define $c_i = \frac{C}{N}$, and the proposal distribution to be

$$
p(\theta, \theta) = \frac{1}{2}, \qquad p(\theta, -\theta) = \frac{1}{2} \qquad \text{for } \theta \in \left\{-\frac{M}{2}, \frac{M}{2}\right\}.
$$

Now, let $0 < q < 1$ be some constant, and consider two cases: (1) $\frac{1}{N}\sum_i x_i = q$ and (2) $\frac{1}{N}\sum_i x_i = -q < 0$. Suppose that in both cases the $x_i$ are shuffled at random. These two cases will have different stationary distributions,

$$\pi_1(\theta) = \frac{1}{Z}\exp\left(-Cq\theta\right) \qquad \text{and} \qquad \pi_2(\theta) = \frac{1}{Z}\exp\left(Cq\theta\right),$$

and an exact algorithm must be able to distinguish between them. Therefore by using these cases, we can get a bound on the required batch size needed for the exact MH algorithm to distinguish between them. First, we observe that the two cases are symmetric, such that if $T_1$ is the transition matrix of the chain in case (1) and $T_2$ is the transition matrix of the chain in case (2), then $T_1(\theta, \theta') = T_2(\theta', \theta)$. Let $0 < \psi < \frac{1}{2}$ denote the probability that $T_1$ transitions from $\frac{M}{2}$ to $-\frac{M}{2}$. Then because the MH method is exact and the chain is reversible, the probability of the reverse transition is $\psi\exp(-CMq)$. So, explicitly, the transition operators will look like

$$T_1 = \begin{bmatrix} 1 - \psi & \psi e^{-CMq} \\ \psi & 1 - \psi e^{-CMq} \end{bmatrix} \qquad \text{and} \qquad T_2 = \begin{bmatrix} 1 - \psi e^{-CMq} & \psi \\ \psi e^{-CMq} & 1 - \psi \end{bmatrix}.$$

The eigenvectors and eigenvalues of this are

$$T_1\pi_1 = \pi_1 \qquad \text{and} \qquad T_1\begin{bmatrix} -1 \\ 1 \end{bmatrix} = (1 - \psi - \psi\exp(-CMq))\begin{bmatrix} -1 \\ 1 \end{bmatrix}.$$

Suppose that we initialize both chains uniformly on $\{-\frac{M}{2}, \frac{M}{2}\}$. Observe that

$$\begin{bmatrix} 1/2 \\ 1/2 \end{bmatrix} = \begin{bmatrix} \frac{\exp(-CMq)}{1+\exp(-CMq)} \\ \frac{1}{1+\exp(-CMq)} \end{bmatrix} + \frac{1 - \exp(-CMq)}{2(1 + \exp(-CMq))}\cdot\begin{bmatrix} 1 \\ -1 \end{bmatrix},$$

the first vector being $\pi_1$ and the second being a multiple of the other eigenvector. Equivalently,

$$\begin{bmatrix} 1/2 \\ 1/2 \end{bmatrix} = \pi_1 + \frac{1}{2}\tanh\left(\frac{CMq}{2}\right)\cdot\begin{bmatrix} 1 \\ -1 \end{bmatrix},$$

and so for any $t$, after $t$ steps of the Markov chain, the distribution will be

$$T_1^t\begin{bmatrix} 1/2 \\ 1/2 \end{bmatrix} = \pi_1 + \frac{1}{2}\tanh\left(\frac{CMq}{2}\right)\cdot(1 - \psi - \psi\exp(-CMq))^t\cdot\begin{bmatrix} 1 \\ -1 \end{bmatrix}.$$

Similarly,

$$T_2^t\begin{bmatrix} 1/2 \\ 1/2 \end{bmatrix} = \pi_2 + \frac{1}{2}\tanh\left(\frac{CMq}{2}\right)\cdot(1 - \psi - \psi\exp(-CMq))^t\cdot\begin{bmatrix} -1 \\ 1 \end{bmatrix}.$$

So, the total variation distance between the state of the chains at time $t$ will be bounded by

$$\mathrm{TV}\left(T_1^t\begin{bmatrix} 1/2 \\ 1/2 \end{bmatrix}, T_2^t\begin{bmatrix} 1/2 \\ 1/2 \end{bmatrix}\right) \geq \mathrm{TV}(\pi_1, \pi_2) - \tanh\left(\frac{CMq}{2}\right)\cdot(1 - \psi - \psi\exp(-CMq))^t.$$

Also observe that

$$\mathrm{TV}(\pi_1, \pi_2) = \frac{1}{2}\left\|\begin{bmatrix} \frac{\exp(-CMq)}{1+\exp(-CMq)} \\ \frac{1}{1+\exp(-CMq)} \end{bmatrix} - \begin{bmatrix} \frac{1}{1+\exp(-CMq)} \\ \frac{\exp(-CMq)}{1+\exp(-CMq)} \end{bmatrix}\right\|_1 = \frac{1 - \exp(-CMq)}{1 + \exp(-CMq)} = \tanh\left(\frac{CMq}{2}\right),$$

so

$$\mathrm{TV}\left(T_1^t\begin{bmatrix} 1/2 \\ 1/2 \end{bmatrix}, T_2^t\begin{bmatrix} 1/2 \\ 1/2 \end{bmatrix}\right) \geq \tanh\left(\frac{CMq}{2}\right)\cdot\left(1 - (1 - \psi - \psi\exp(-CMq))^t\right).$$

Also, since we know that our algorithm is guaranteed to have spectral gap ratio at least $\kappa$ with the original chain, it follows that $\psi \geq \kappa/2$, and so

$$\mathrm{TV}\left(T_1^t\begin{bmatrix} 1/2 \\ 1/2 \end{bmatrix}, T_2^t\begin{bmatrix} 1/2 \\ 1/2 \end{bmatrix}\right) \geq \tanh\left(\frac{CMq}{2}\right)\cdot\left(1 - \left(1 - \frac{\kappa}{2} - \frac{\kappa}{2}\exp(-CMq)\right)^t\right).$$

Now, denote the exact minibatch algorithm to be $\mathcal{A}$. As it runs, the algorithm $\mathcal{A}$ will request data examples by querying the energy difference oracle. Under case (1), we let $y_i$ denote the $i$th sample

that $\mathcal{A}$ *would have observed* if it requested $i$ or more samples, and similarly we let $z_i$ denote the analogous sample in case (2). Fix some constant $t \in \mathbf{N}$ (which we will set later). We let $K_1$ denote the total number of samples observed by $\mathcal{A}$ across the first $t$ iterations in case (1), and set

$$\mu = \{y_1, y_2, \dots, y_{K_1}\}.$$

Similarly, we let $K_2$ denote the number of samples observed by $\mathcal{A}$ across the first $t$ iterations in case (2), and set

$$\nu = \{z_1, z_2, \dots, z_{K_2}\}.$$

Now, we fix some constant $K$ (to be set later), and consider the following coupling between the behavior of $\mathcal{A}$ across its first $t$ iterations in case (1) and in case (2). First, let all internal randomness of $\mathcal{A}$ and the proposal process under case (1) and (2) be the same, which means that for a given observation of data examples, the algorithm $\mathcal{A}$ will make the same decision, such as whether to require more data examples or not and whether to accept or not. Second, choose a coupling that minimizes the probability that

$$(y_1, y_2, \dots, y_{K1}) \neq (z_1, z_2, \dots, z_{K2}).$$

Such a coupling is guaranteed to exist by the Coupling Lemma, and the probability that these two are not equal will be equal to the total variation distance between their distributions. Third, assign all the other $y_i$ and $z_i$, for $i > K$, independently according to their distribution.

We are interested in the quantity $p(\mu \neq \nu)$, which bounds the probability that the algorithm may make a different decision in cases (1) and (2). We can decompose this probability into two terms,

$$p(\mu \neq \nu) = p(\mu \neq \nu \text{ and } y_j = z_j \text{ for all } j \leq K) + p(\mu \neq \nu \text{ and } y_j \neq z_j \text{ for some } j \leq K).$$

If $\mu \neq \nu$ but $y_j = z_j$ for all $j \leq K$, the only way that this is possible is for $K_1 > K$ (and, symmetrically, also $K_2 > K$), since otherwise the algorithms would behave identically. So,

$$p(\mu \neq \nu) \leq p(K_1 > K) + p(y_j \neq z_j \text{ for some } j \leq K). \tag{10}$$

By Markov's inequality,

$$p(\mu \neq \nu) \leq \frac{\mathbf{E}[K_1]}{K} + p(y_j \neq z_j \text{ for some } j \leq K).$$

For the second term of (10), we can reduce the case to only considering $K$ samples. Let $S_y$ be the total number of samples $y_i$ that are $-1$ and let $S_z$ be the total number of samples $z_i$ that are $-1$. Since $\mathcal{A}$ is effectively sampling a shuffled dataset at some arbitrary indices without replacement, both of these random variables $S_y$ and $S_z$ are—properly speaking—hypergeometric random variables. However, since our dataset size $N$ is arbitrary here, we can by setting $N$ very large work in the limit (as $N \to \infty$) in which these variables become binomial (since sampling with replacement and without replacement can be made to have arbitrarily close to the same distribution by making the dataset large). Observe that (in this limit) $S_y$ follows a binomial distribution $B(K, \frac{1-q}{2})$ and $S_z$ follows a binomial distribution $B(K, \frac{1+q}{2})$. Clearly, if $S_y = S_z$, then we can arrange the coupling so that $(y_1, \dots, y_K) = (z_1, \dots, z_K)$. So, by the Coupling Lemma,

$$p(y_j \neq z_j \text{ for some } j \leq K) = p(S_y \neq S_z) = \text{TV}(S_y, S_z).$$

From the analysis in Adell and Jodrá [1], we can bound the total variance distance between these two binomial variables with

$$\text{TV}(S_y, S_z) \leq \sqrt{e} \cdot \frac{\tau}{(1-\tau)^2}$$

where $\tau = \sqrt{\frac{K+2}{2}} \cdot q < 1$. Substituting these bounds, we get

$$p(\mu \neq \nu) \leq \frac{\mathbf{E}[K_1]}{K} + \sqrt{e} \cdot \frac{\tau}{(1-\tau)^2}.$$

But the probability that $\mu \neq \nu$ must be an upper bound on the probability that the distributions of the chains in case (1) and (2) after $t$ steps are not equal, since if $\mu = \nu$ in the coupling then the two chains are in the same state. So, using our bound from earlier, we get

$$\tanh\left(\frac{CMq}{2}\right) \cdot \left(1 - \left(1 - \frac{1}{2}\kappa - \frac{1}{2}\kappa \exp(-CMq)\right)^t\right) \leq \frac{\mathbf{E}[K_1]}{K} + \sqrt{e} \cdot \frac{\tau}{(1-\tau)^2}.$$

Now isolating $\mathbf{E}[K_1]$ gives

$$K \cdot \tanh\left(\frac{CMq}{2}\right) \cdot \left(1 - \left(1 - \frac{1}{2}\kappa - \frac{1}{2}\kappa \exp(-CMq)\right)^t\right) - K \cdot \sqrt{e} \cdot \frac{\tau}{(1-\tau)^2} \leq \mathbf{E}[K_1].$$

Also, observe that

$$\left(1 - \frac{1}{2}\kappa - \frac{1}{2}\kappa \exp(-CMq)\right)^t \leq \left(1 - \frac{1}{2}\kappa\right)^t \leq \exp\left(-\frac{\kappa t}{2}\right),$$

so

$$K \cdot \tanh\left(\frac{CMq}{2}\right) \cdot \left(1 - \exp\left(-\frac{\kappa t}{2}\right)\right) - K \cdot \sqrt{e} \cdot \frac{\tau}{(1-\tau)^2} \leq \mathbf{E}[K_1].$$

This gives us the lower bound on $\mathbf{E}[K_1]$ that we are interested in. Now, it remains to assign $q$, $K$, and $t$. We start by assigning $t$ such that

$$t = \left\lceil 2\kappa^{-1}\log(2) \right\rceil,$$

in which case

$$\exp\left(-\frac{\kappa t}{2}\right) \leq \frac{1}{2}$$

and so

$$K \cdot \frac{1}{2} \cdot \tanh\left(\frac{CMq}{2}\right) - K \cdot \sqrt{e} \cdot \frac{\tau}{(1-\tau)^2} \leq \mathbf{E}[K_1].$$

Now, we add some simplifying assumptions, which we will validate are true later. We assume that

$$\tau = \sqrt{\frac{K+2}{2}} \cdot q \leq \frac{1}{2};$$

in this case

$$\sqrt{e} \cdot \frac{\tau}{(1-\tau)^2} \cdot K \leq 4\sqrt{e} \cdot \tau \leq 5\sqrt{K+2} \cdot q.$$

We set $q$ such that

$$CMq = 1,$$

and we assume that $CM$ is large enough that this assignment of $q$ is within range (i.e. $0 < q < 1$). This gives us

$$K \cdot \frac{1}{2} \cdot \tanh\left(\frac{1}{2}\right) - 5K\sqrt{K+2} \cdot \frac{1}{CM} \leq \mathbf{E}[K_1].$$

Since $\tanh(1/2) > 5/16$, we can simplify this to

$$K \cdot \frac{5}{32} - 5K\sqrt{K+2} \cdot \frac{1}{CM} \leq \mathbf{E}[K_1].$$

All that remains is to assign $K$. We assign $K$ such that

$$\sqrt{K+2} \cdot \frac{1}{CM} = \frac{1}{64}.$$

In this case, we get

$$K = \frac{C^2 M^2}{4096} - 2,$$

and our bound reduces to

$$\left(\frac{C^2 M^2}{4096} - 2\right) \cdot \frac{5}{64} \leq \mathbf{E}[K_1].$$

We can simplify this further to

$$2^{-16} \cdot C^2 M^2 - \frac{5}{32} \leq \mathbf{E}[K_1].$$

Now, this is a bound on the expected number of samples taken across $t$ iterations. This means that the number of samples taken in any given iteration will be bounded by

$$\frac{\mathbf{E}[K_1]}{t} \geq \frac{2^{-16} \cdot C^2 M^2 - \frac{5}{32}}{2\kappa^{-1}\log(2) + 1} = \frac{2^{-16} \cdot \kappa C^2 M^2 - \frac{5\kappa}{32}}{2\log(2) + \kappa}.$$

A few more loose bounds, leveraging $\kappa < 1$, gives us

$$\frac{\mathbf{E}[K_1]}{t} \geq 2^{-18} \cdot \kappa C^2 M^2 - \frac{\kappa}{16}.$$

This proves the lemma. □

Next, we will show the following lemma, which characterizes what happens when $CM$ is small.

**Lemma 2.** *Considering minibatch MH algorithms in the same setting as the theorem, the expected batch size at any iteration must be lower bounded by*

$$\mathbf{E}[B] \geq \frac{\kappa}{2} \min\left(CM(\theta, \theta'), 1\right).$$

*Proof.* Here, we will prove a lower bound that characterizes the limits of exact stateless minibatch MH algorithms when they use very few examples. Again, without loss of generality we consider a reduction to the two-state case as we did in the proof of the previous lemma. Suppose that a exact stateless minibatch MH algorithm with the same forward and backward proposal probabilities (given some $c_1, \ldots, c_N$, $C$, and $M$) requests any energy function examples at all only with probability $p$. Consider two cases, which have the same $c_1, \ldots, c_N$, $C$ and $M$. In the first case,

$$\sum_{i=1}^{n}(U_i(\theta) - U_i(\theta')) = CM(\theta, \theta'),$$

while in the second case,

$$\sum_{i=1}^{n}(U_i(\theta) - U_i(\theta')) = -CM(\theta, \theta').$$

These are clearly possible by setting $U_i$ to the limits of what is covered by the bounds. In the first case, the baseline MH method would accept with probability 1. In the second case, it will accept with probability $\exp(-CM(\theta, \theta'))$. Since the stateless MH algorithm is reversible, it must accept in the first case with some probability $a$ and in the second case with probability $a \cdot \exp(-CM(\theta, \theta'))$. But, the algorithm can only distinguish the two cases if it requests samples, which only happens with probability at most $p$. So,

$$a - a \cdot \exp(-CM(\theta, \theta') \leq p.$$

Since we know that it must be the case that $a \geq \kappa$ (from a straightforward analysis of a two-state case), it follows that

$$\frac{p}{\kappa} \geq \frac{p}{a} \geq 1 - \exp(-CM(\theta, \theta')) \geq \frac{1}{2} \min\left(CM(\theta, \theta'), 1\right).$$

Since $p$ is an obvious lower bound on the expected value of the batch size, it follows that

$$\mathbf{E}[B] \geq \frac{\kappa}{2} \min\left(CM(\theta, \theta'), 1\right).$$

□

To prove Theorem 3 we now combine the results of these two lemmas. We have

$$\mathbf{E}[B] \geq 2^{-18} \cdot \kappa C^2 M^2(\theta, \theta') - 2^{-4} \cdot \kappa.$$

and

$$\mathbf{E}[B] \geq \frac{\kappa}{2} \min\left(CM(\theta, \theta'), 1\right).$$

Since these are both lower bounds, we can combine them to get

$$\mathbf{E}[B] \geq \max\left(2^{-18} \cdot \kappa C^2 M^2(\theta, \theta') - 2^{-4} \cdot \kappa, \frac{\kappa}{2} \min\left(CM(\theta, \theta'), 1\right)\right)$$

$$= \kappa \cdot \max\left(2^{-18} \cdot C^2 M^2(\theta, \theta') - 2^{-4}, \frac{1}{2} \min\left(CM(\theta, \theta'), 1\right)\right).$$

It is obvious from a simple big-$\mathcal{O}$ analysis here that there exists a global constant $\zeta > 0$ such that

$$\mathbf{E}[B] \geq \zeta \cdot \kappa \left(C^2 M^2(\theta, \theta') + CM(\theta, \theta')\right).$$

This proves the theorem.

# I   Proof of Corollary 1

*Proof.* Recall that the lower bound on the batch size in each iteration is

$$\mathbf{E}[B] \geq \zeta \cdot \kappa \left( C^2 M^2(\theta, \theta') + CM(\theta, \theta') \right).$$

Since $C = \mathcal{O}(N)$ and $M(\theta, \theta') = \mathcal{O}(N^{-(h+1)/2})$, the expectation of the batch size follows

$$\mathbf{E}[B] = \mathcal{O}(C^2 M^2(\theta, \theta') + CM(\theta, \theta')) = \mathcal{O}(CM(\theta, \theta')) = \mathcal{O}(N^{1-h}/2).$$

When $h = 1$, $\mathbf{E}[B] = \mathcal{O}(1)$ and when $h = 2$, $\mathbf{E}[B] = \mathcal{O}(1/\sqrt{N})$.   □

## J   Experimental Details and Additional Results

### J.1   Experiment in Section 2.1

To verify Theorem 1, we empirically construct a distribution in the form of Section A such that AustereMH and MHminibatch are biased on. Note that the proof in Section A shows there must exist such a distribution for any inexact minibatch method but does not tell us how to find one for a specific method. Therefore, in order to find such a distribution, we construct an example and empirically test whether AustereMH and MHminibatch are biased on it.

We let data $x_i$ take one of two values $\{-1, 5\}$. Consider a dataset of size 6000. We let 5000 data take value $-1$ and the remaining 1000 data take value 5. Define the target distribution $\pi(\theta)$ to be

$$\pi(\theta) \propto \exp\left( -\frac{1}{N} \sum_{i=1}^{N} \theta \cdot x_i \right)$$

where the domain of $\theta$ is $\{0, 1, \ldots, K-1\}$. Therefore the number of state is $K$. Since $\sum_i x_i = 0$, it is clear to see that the stationary distribution of $\theta$ is a uniform distribution. We define the proposal distribution to be the following

$$p(\theta, \theta) = \frac{1}{2}, \quad \text{for all } \theta; \quad p(\theta, \theta - 1) = \frac{1}{4}, \quad p(\theta, \theta + 1) = \frac{1}{4} \quad \text{for } \theta \in \{1, \ldots, K-2\};$$

and $p(0, 1) = p(K-1, K-2) = \frac{1}{2}$.

We set the hyperparameter error $\epsilon$ in AustereMH to be 0.01 and $\delta$ in MHminibatch to be 5, following the setting in their original papers [17, 26]. We set batch size $m$ in both methods to be 30. We find that AustereMH and MHminibatch are both inexact on this example and the error increases as we increase $K$. Thus we empirically verify the statement in Theorem 1.

Besides the density estimate comparison on $K = 200$ shown in Figure 1b, we additionally report the estimate results on other values of $K$ in Figure 5. We see that the results are similar, all showing that TunaMH and standard MH can give accurate estimate whereas inexact methods are seriously wrong.

**On Robust Linear Regression**   We further tested AustereMH on robust linear regression in Section 5.1 with $N = 5000$. We computed the MSE between estimated and true parameters. MH, TunaMH and AustereMH obtained MSE 0.149, 0.15 and 1.19 respectively, indicating inexact method error can be large on typical problems.

### J.2   Robust Linear Regression

We follow the experimental setup of robust linear regression (RLR) in Cornish et al. [12]. Specifically, we have data $x_i \in \mathbb{R}^d$ and $y_i \in \mathbb{R}$. The likelihood is modeled by a student's t-distribution with degrees of freedom $v$:

$$p(y_i|\theta, x_i) = \text{Student}(y_i - \theta^\mathsf{T} x_i | v).$$

It follows that

$$U_i(\theta) = \frac{v+1}{2} \log\left( 1 + \frac{(y_i - \theta^\mathsf{T} x_i)^2}{v} \right),$$

(a) $K = 500$  (b) $K = 1000$

(e) $K = 2000$  (e) $K = 5000$

Figure 5: Density estimate comparison on $K = 500, 1000, 2000, 5000$.

and the first derivative

$$\partial_j U_i(\theta) = -(v+1)\frac{x_{ij}(y_i - \theta^\mathsf{T} x_i)}{v + (y_i - \theta^\mathsf{T} x_i)^2}.$$

Since the function $U_i$ is Lipschitz continuous, we can easily get the bound used in TunaMH, TFMH and SMH. We set $M(\theta, \theta') = \|\theta - \theta'\|_2$ and then it follows

$$c_i = \sup_{\theta \in \mathbb{R}} \|\nabla U_i(\theta)\|_2 = \frac{v+1}{2\sqrt{v}} \|x_i\|_2.$$

The data $x_i$ and $y_i$ is generated as follows

$$y_i = \sum_j x_{ij} + \epsilon_i$$

where $\epsilon_i \sim \mathcal{N}(0, 1)$.

In Section 5.1, we set $v = 4$, $d = 100$ and use a flat prior $p(\theta) = 1$. Note that our problem dimension $d$ is much larger than that in the SMH paper [12] ($d = 10$). This makes the control variates in SMH problematic since the bounds they require appear to scale badly in high dimensions.

To reach the target acceptance rate, we set the stepsize in each method as in Table 2 and 3. For TunaMH and TunaMH-MAP, we set $\chi = 1e - 5$ for $N = 5000, 20000$ and $\chi = 1e - 4$ for $N = 50000, 100000$. For FlyMC and FlyMC-MAP, we set the probability for a data going from dark to bright $q_{d \to b}$ to be 0.01. Without the MAP, we collect 80000 samples after 200000 step burnin. With the MAP, we collect 80000 samples without burnin.

Table 2: Stepsize of methods without the MAP.

|  | MH | TFMH | FlyMC | TunaMH |
|---|---|---|---|---|
| RLR $N = 5000$ | 4e-3 | 1e-4 | 2.7e-3 | 8e-4, $\chi = 1e - 5$ |
| RLR $N = 20000$ | 2e-3 | 3e-5 | 1.5e-3 | 3e-4, $\chi = 1e - 5$ |
| RLR $N = 50000$ | 1.3e-3 | 1.2e-5 | 9e-4 | 2e-4, $\chi = 1e - 4$ |
| RLR $N = 100000$ | 9e-4 | 6e-6 | 7e-4 | 1.7e-4, $\chi = 1e - 4$ |
| TGM | 3e-1 | 2.2e-2 | 1e-2 | 1e-1 |
| LR | 5e-3 | 1e-4 | 2e-3 | 1e-3 |

Table 3: Stepsize of methods with the MAP.

| | MH-MAP | SMH-1 | SMH-2 | FlyMC-MAP | TunaMH-MAP |
|---|---|---|---|---|---|
| RLR $N = 5000$ | 4e-3 | 4e-3 | 4e-3 | 6e-3 | 8e-4, $\chi = 1e-5$ |
| RLR $N = 20000$ | 2e-3 | 2e-3 | 2e-3 | 3.5e-3 | 3e-4, $\chi = 1e-5$ |
| RLR $N = 50000$ | 1.2e-3 | 1.2e-3 | 1.2e-3 | 2.5e-3 | 1.2e-4, $\chi = 1e-4$ |
| RLR $N = 100000$ | 9e-4 | 5.9e-4 | 8e-4 | 1.7e-3 | 7e-5. $\chi = 1e-4$ |
| TGM | - | 1e-1 | - | 1e-2 | - |

### J.2.1  Additional Experimental Results with $d = 10$

We ran RLR experiment with $d = 10$ and $N = 10^5$ to compare the performance in low dimensions. The ESS/S for TFMH, FlyMC, TunaMH are 0.02, 0.75, & 1.7, respectively; SMH-1, SMH-2, FlyMC-MAP and TunaMH-MAP are 174.7, 5969.5, 730.8, & 730.1 respectively. This suggests TunaMH is significantly better without MAP/control variates. With MAP/control variates, TunaMH is better than SMH-1, similar to FlyMC and worse than SMH-2.

### J.3  Truncated Gaussian Mixture

The data in this truncated Gaussian mixture (TGM) task is generated as follows

$$x_i \sim \frac{1}{2}\mathcal{N}(\theta_1, \sigma_x^2) + \frac{1}{2}\mathcal{N}(\theta_1 + \theta_2, \sigma_x^2)$$

where $\theta_1 = 0, \theta_2 = 1$ and $\sigma^2 = 2$. The posterior $\theta$ has two modes at $(\theta_1, \theta_2) = (0, 1)$ and $(\theta_1, \theta_2) = (1, -1)$. In order to get the bounds required by all methods, we truncate the Gaussian by setting $\theta_1, \theta_2 \in [-3, 3]$.

For simplicity we assume a flat prior $p(\theta) = 1$. Then the energy is given by

$$U_i(\theta) = -\log p(x_i|\theta) = \log(2\sqrt{2\pi}\sigma_x) - \log\left[\exp\left(-\frac{(x_i - \theta_1)^2}{2\sigma_x^2}\right) + \exp\left(-\frac{(x_i - \theta_1 - \theta_2)^2}{2\sigma_x^2}\right)\right].$$

Denote $E_1 = \exp\left(-\frac{(x_i-\theta_1)^2}{2\sigma_x^2}\right)$ and $E_2 = \exp\left(-\frac{(x_i-\theta_1-\theta_2)^2}{2\sigma_x^2}\right)$. To get the upper bound in TunaMH, TFMH and SMH, we compute the gradient

$$\frac{\partial U_i(\theta)}{\partial \theta_1} = -\frac{1}{E_1 + E_2}\left(E_1 \cdot \frac{x_i - \theta_1}{\sigma_x^2} + E_2 \cdot \frac{x_i - \theta_1 - \theta_2}{\sigma_x^2}\right),$$

$$\frac{\partial U_i(\theta)}{\partial \theta_2} = -\frac{1}{E_1 + E_2}\left(E_2 \cdot \frac{x_i - \theta_1 - \theta_2}{\sigma_x^2}\right).$$

Since $\theta_i \in [-3, 3]$, it follows that

$$\left|\frac{\partial U_i(\theta)}{\partial \theta_1}\right| \leq \frac{|x_i| + 3}{\sigma_x^2} + \frac{|x_i| + 3 + 3}{\sigma_x^2} \leq \frac{2|x_i| + 9}{\sigma_x^2},$$

$$\left|\frac{\partial U_i(\theta)}{\partial \theta_2}\right| \leq \frac{|x_i| + 3 + 3}{\sigma_x^2} \leq \frac{|x_i| + 6}{\sigma_x^2}.$$

Therefore we can set $M(\theta, \theta') = \|\theta - \theta'\|_2$ and

$$c_i = \sqrt{\left(\frac{2|x_i| + 9}{\sigma_x^2}\right)^2 + \left(\frac{|x_i| + 6}{\sigma_x^2}\right)^2}.$$

To use the control variate in SMH, we need to compute the second derivatives

$$\frac{\partial^2 U_i(\theta)}{\partial^2 \theta_1} = \frac{1}{(E_1 + E_2)^2} \cdot \left( E_1 \cdot \frac{x_i - \theta_1}{\sigma_x^2} + E_2 \cdot \frac{x_i - \theta_1 - \theta_2}{\sigma_x^2} \right)^2$$
$$- \left[ E_1 \cdot \left( \left( \frac{x_i - \theta_1}{\sigma_x^2} \right)^2 - \frac{1}{\sigma_x^2} \right) + E_2 \cdot \left( \left( \frac{x_i - \theta_1 - \theta_2}{\sigma_x^2} \right)^2 - \frac{1}{\sigma_x^2} \right) \right] \cdot \frac{1}{E_1 + E_2}$$

$$\frac{\partial^2 U_i(\theta)}{\partial \theta_1 \partial \theta_2} = \frac{1}{(E_1 + E_2)^2} \cdot \left( E_2 \cdot \left( \frac{x_i - \theta_1 - \theta_2}{\sigma_x^2} \right) \right) \cdot \left( E_1 \cdot \frac{x_i - \theta_1}{\sigma_x^2} + E_2 \cdot \frac{x_i - \theta_1 - \theta_2}{\sigma_x^2} \right)$$
$$- \left[ E_2 \left( \left( \frac{x_i - \theta_1 - \theta_2}{\sigma_x^2} \right)^2 - \frac{1}{\sigma_x^2} \right) \right] \cdot \frac{1}{E_1 + E_2}$$

$$\frac{\partial^2 U_i(\theta)}{\partial^2 \theta_2} = \frac{1}{(E_1 + E_2)^2} \cdot \left( E_1 \cdot \frac{x_i - \theta_1}{\sigma_x^2} + E_2 \cdot \frac{x_i - \theta_1 - \theta_2}{\sigma_x^2} \right)^2$$
$$- \left[ E_2 \cdot \left( \left( \frac{x_i - \theta_1 - \theta_2}{\sigma_x^2} \right)^2 - \frac{1}{\sigma_x^2} \right) \right] \cdot \frac{1}{E_1 + E_2}.$$

Given the parameter space, we have the upper bounds

$$\left| \frac{\partial^2 U_i(\theta)}{\partial^2 \theta_1} \right| \leq \left( \frac{2|x_i| + 9}{\sigma_x^2} \right)^2 + \left( \frac{|x_i| + 3}{\sigma_x^2} \right)^2 + \left( \frac{|x_i| + 6}{\sigma_x^2} \right)^2 + \frac{2}{\sigma_x^2}$$

$$\left| \frac{\partial^2 U_i(\theta)}{\partial \theta_1 \partial \theta_2} \right| \leq \frac{2|x_i| + 9}{\sigma_x^2} \cdot \frac{|x_i| + 6}{\sigma_x^2} + \left( \frac{|x_i| + 6}{\sigma_x^2} \right)^2 + \frac{1}{\sigma_x^2}$$

$$\left| \frac{\partial^2 U_i(\theta)}{\partial^2 \theta_2} \right| \leq \left( \frac{2|x_i| + 9}{\sigma_x^2} \right)^2 + \left( \frac{|x_i| + 6}{\sigma_x^2} \right)^2 + \frac{1}{\sigma_x^2}.$$

It follows

$$\bar{U}_{2,i} = \left( \frac{2|x_i| + 9}{\sigma_x^2} \right)^2 + \left( \frac{|x_i| + 3}{\sigma_x^2} \right)^2 + \left( \frac{|x_i| + 6}{\sigma_x^2} \right)^2 + \frac{2}{\sigma_x^2}.$$

which is required in SMH-1.

To get the lower bounds in FlyMC, we use the first-order Taylor expansion for $U_i(\theta)$. Higher order approximation is possible but would require heavier computation. By Taylor expansion,

$$U_i(\theta) = U_i(\theta^0) + \nabla U_i(\theta^0)^\intercal (\theta - \theta^0) + \frac{1}{2}(\theta - \theta^0)^\intercal \nabla^2 U_i(c)(\theta - \theta^0)$$

where $c$ is between $\theta$ and $\theta^0$.

Then we can define $\log B_i(\theta)$ in FlyMC as the follows

$$\log B_i(\theta) = -U_i(\theta^0) - \nabla U_i(\theta^0)^\intercal (\theta - \theta^0) - \frac{1}{2} \cdot \max_c \left\| \nabla^2 U_i(c) \right\|_1 \cdot \left\| \theta - \theta^0 \right\|_1^2$$
$$= -U_i(\theta^0) - \nabla U_i(\theta^0)^\intercal (\theta - \theta^0) - \frac{1}{2} \cdot \bar{U}_{2,i} \cdot \left\| \theta - \theta^0 \right\|_1^2.$$

The sum of $\log B_i$ is

$$\sum_{i=1}^N \log B_i(\theta) = -N \cdot U_i(\theta^0) - \left( \sum_{i=1}^N \nabla U_i(\theta^0) \right)^\intercal (\theta - \theta^0) - \frac{1}{2} \cdot \sum_{i=1}^N \bar{U}_{2,i} \cdot \left\| \theta - \theta^0 \right\|_1^2.$$

We set $\theta^0$ to be 0 and the MAP solution in standard and MAP-tuned FlyMC respectively.

We tune the stepsize of each method to reach the acceptance rate 60% and the value of stepsize is summarized in Table 2 and 3. We set $\chi = 10^{-4}$ in TunaMH and $q_{d \to b} = 0.01$ in FlyMC and FlyMC-MAP. We compute the symmetric KL between the run-average density estimate and the true distribution. Since this is a two-dimensional problem, we are able to visualize the density estimate. As shown in Figure 6, we plot the density estimate after running the method for 1 second. It is clear to see that the density estimate of TunaMH is close to the truth whereas all other methods are unable to provide accurate density estimate given the time budget.

Figure 6: Visualization of the density estimate after 1 second.

## J.4 Logistic Regression on MNIST

MNIST with only 7s and 9s images contains 12214 training data and 2037 test data. Let $h$ be the sigmoid function. Let the label $y_i \in \{0, 1\}$, then the model in logistic regression (LR) is

$$p(y_i = 1) = h(\theta^\mathsf{T} x_i) = \frac{1}{1 + \exp\left(-\theta^\mathsf{T} x_i\right)}.$$

It follows that

$$U_i(\theta) = -y_i \log h\left(\theta^\mathsf{T} x_i\right) - (1 - y_i) \log h\left(-\theta^\mathsf{T} x_i\right).$$

It is easy to see that

$$|\partial_j U_i| = |(h(\theta^\mathsf{T} x_i) - y_i)x_{ij}| \leq 1 \cdot |x_{ij}|.$$

Thus we can set $M(\theta, \theta')$ to be $\|\theta - \theta'\|_2$ and $c_i$ to be $\|x_i\|_2$. We use this bound for TunaMH, TFMH and SMH. For FlyMC, we use the same bound on logistic regression as in the FlyMC paper [20].

We set the target acceptance rate to be $60\%$ and the resulted stepsize is reported in Table 2. We set $q_{d \to b}$ to be 0.1 following [20].