[Reviews · NeurIPS 2020]

Review 1

Summary and Contributions: This paper is a study of “minibatch MH” algorithms: variants of Metropolis-Hastings that consider only a subset of factors of a probabilistic model’s target density when deciding whether to accept or reject a proposal. It divides these methods into two categories: inexact methods, which are not guaranteed to converge to the correct distribution, and exact methods, which do target the correct distribution but may require more iterations to converge. The paper is concerned only with stateless methods, in which only the current parameter theta is considered when making a proposal and deciding to accept or reject – no additional algorithm-specific state is tracked. The authors make several contributions: • The authors prove theorems about the worst-case behavior of inexact algorithms and existing exact algorithms, by constructing adversarial target distributions and proposals on which these algorithms perform poorly (large bias in inexact algorithms, and slow convergence or large minibatches for some existing exact algorithms). • The authors present a new exact method, TunaMH, for minibatch MH. The method is similar to PoissonMH. At each iteration, the state space is temporarily extended with Poisson random variables whose rates are carefully constructed so that the extended target density does not depend on all factors of the original target. A standard Metropolis-Hastings move is then performed in this extended space before forgetting the extra Poisson variables (so they are sampled anew at the next iteration). The innovation in TunaMH is that the rate parameter for the Poisson variables can depend not only on the current parameter value theta, but also on the proposed value theta’. Knowing theta and theta’, it is often possible (thanks to, e.g., Lipschitz continuity of the target energy function) to bound the maximum change in energy on a move-by-move basis, more tightly than would be possible using PoissonMH’s global bounds. These tighter bounds allow for smaller Poisson rates, and ultimately smaller minibatches (since a factor is included in the minibatch whenever its corresponding Poisson variable is greater than 0). • The authors demonstrate empirically that in 100-dimensional robust linear regression (100,000 data points), logistic regression on MNIST (distinguishing between 7s and 9s, using 50 features), and a two-component truncated Gaussian mixture model (with 1,000,000 data points), TunaMH outperforms other exact ‘minibatch MH’ algorithms, according to various metrics (ESS/second for linear regression, test accuracy for logistic regression, and symmetric KL for GMM).

Strengths: This is a well-written paper that provides an interesting study of the space of minibatch MH algorithms. I found the examples developed in the constructive proofs to be enlightening. Furthermore, the TunaMH algorithm seems to be a valuable improvement over the state of the art. MH has, of course, been applied to solve a huge variety of problems across many disciplines; it is of intense interest (to the NeurIPS community and others) to develop methods that can help MH scale to larger datasets and more expensive likelihoods. I think many in the community would appreciate this interesting paper, and the algorithm looks simple enough to implement, and broadly applicable enough, that it could find many users.

Weaknesses: One weakness, in my view, is that there is no empirical comparison with inexact methods on real-world datasets. (Figure 1 is a nice illustration, but if I understand correctly, it uses an adversarially constructed target and proposal designed to foil the inexact algorithms. How does TunaMH compare in practice to these inexact method, on typical problems?) Another experiment I would have loved to see is the application of TunaMH to a more difficult problem (i.e., where convergence takes longer than a few seconds) and/or to a more sophisticated Metropolis-Hastings algorithm, e.g., a split/merge reversible-jump algorithm for change-point analysis, or a sampler for a non-parametric model with non-conjugate priors, or perhaps even some sort of particle Gibbs sampler in a state-space model (though I realize it is possible that none of these is especially straight-forward). The point is just that for problems like logistic regression, we have plenty of methods for handling big data, whereas there are many harder problems that Metropolis-Hastings can solve in ‘small data’ regimes but for which solutions not based on Monte Carlo are scarce. It would be awesome to see evidence that a method like TunaMH could help scale the Monte Carlo approach up.

Correctness: Some of the proofs (e.g. about the spectral gap) I have not checked in complete detail, but as far as I can tell, the results presented here are correct (the algorithm really is stationary for the target posterior, etc.).

Clarity: I found the paper well-written and relatively easy to read. [One small point of confusion for me was that no conditions are placed on M(theta, theta’). At first I wondered why one couldn’t simply set M(theta, theta’) = max_i |U_i(theta) – U_i(theta’)|. Of course, this is silly – this would require computing all of U at every iteration in order to compute M(theta, theta’), completely defeating the purpose of the algorithm. But I wonder if some intuition might be provided here that the real restriction is that M be simple / fast to compute, and not depend on the details of all the U_i. Maybe this confusion was just me.]

Relation to Prior Work: I believe so, though I am not an expert in minibatch MH algorithms, and would not necessarily be aware if there were something missing.

Reproducibility: Yes

Additional Feedback: Post response: =========== Thank you to the authors for your response; I enjoyed reading it and appreciated your sharing of empirical results for the AustereMH baseline on robust linear regression. I reaffirm my recommendation to accept this paper. One small comment: I do agree with R5’s point, under “Clarity,” that the derivation presented in Appendix C is a little bit misleading, as in PoissonMH, and in many other auxiliary variable schemes for MH (such as Hamiltonian Monte Carlo), it is the case that the extended target does not depend on the location of the MH proposal. It is not immediately obvious, reading Appendix C, why it is OK that TunaMH’s extended target does depend on theta’. I believe, but not have not carefully checked, that one can understand the TunaMH algorithm by considering both theta and theta’ to be part of the extended target distribution (which also includes the proposal density q). At each iteration, theta’ and the other auxiliary variables are Gibbs-sampled in a block, conditioning on theta, and then a deterministic MH proposal is made to *swap* theta and theta’ (leading to the acceptance probability you derive). If accepted, this has the effect of updating theta to the proposed theta’. Otherwise, theta’ will be overwritten at the next iteration (and thus can be safely forgotten).


Review 2

Summary and Contributions: This paper studies Metropolis-Hastings algorithms for posterior distributions, with only a minibatch of the dataset queried at each iteration and the exact distribution preserved. The paper first illustrates the need for exact sampling methods, showing that when the sampled distribution is inexact, there exist target distributions that give arbitrarily bad results. A new algorithm is proposed and is it shown to exhibits a certain optimality property. It includes some illustrative experiments as well. I found the ideas well motivated, well executed and clearly written.

Strengths: The paper is theoretically sound and the empirical evaluation compelling

Weaknesses: It seems that there is one loose end in the results presented: how does the expected batch size E[B] compare to N? Since E[B]> C*M(\theta,\theta'), and C is O(N), it seems possible that E[B] > N. Any comments on that case? Typos: - in Algorithm 2, in the denominator of the acceptance probability for sample i_b, there is c_i which should be c_{i_b}. - In line 575 (appendix), one of the A's should be a B.

Correctness: I read the proofs in the appendix, and while I did not check every detail, they seem correct.

Clarity: Yes, the presentation is clear and well organized.

Relation to Prior Work: Yes, the paper provides the right context to appreciate the new contribution.

Reproducibility: Yes

Additional Feedback:


Review 3

Summary and Contributions: EDIT: I've read the rebuttal and I am satisfied with the answers. Metropolis-Hastings becomes impractical when the number N of data items is large, and the quest for a provably accurate MH algorithm that uses as few data as possible at each iteration has been ongoing. The authors propose an original "theoretical computer science" approach to the problem, 1) showing that any inexact subsampling variant of MH can fail arbitrarily badly on a suitably chosen target and proposal, 2) establishing a "worst case" lower bound on the size of the batchsize, and 3) giving an algorithm that reaches this gets close to this lower bound, and provably doesn't loose too much with respect to the underlying intractable MH in terms of spectral gap.

Strengths: The topic is impactful, the paper clearly written, and the approach interesting. In particular, I find it refreshing to see a different approach to subsampling MH, which I loosely term "theoretical CS"-minded. Even if the impact of the proposed theoretical results can be discussed, in particular the rather adhoc worst case constructions in the proofs, results of the kind of Theorem 3 are novel in the subsampling literature as far as I know. I believe they provide food for thought and would be a great topic of discussion at NeurIPS. Moreover, the proposed algorithm is rather simple and shows promising empirical performance, even when compared to approaches using Taylor-based control variates.

Weaknesses: -My concerns are mostly about 1) the need to justify some strong comments on prior work and 2) the representability of the worst case constructions in the proofs; see main comments below.

Correctness: seems correct, although I wasn't able to proofread all proofs in the given time.

Clarity: yes

Relation to Prior Work: yes

Reproducibility: Yes

Additional Feedback: # Major comments * L82: maybe give here one or two examples of U, C, and M, say for logistic regression and a more complex example. * Theorem 1: I appreciate the negative result, and I like to see a maybe more TCS approach to Bayes a big data. That being said, I am a bit puzzled by the statement "For a given algorithm there exists a target and a proposal" of the theorem and the construction in the proof. Maybe by taste, if I were to make an argument against an inexact algorithm, I would rather design a nonpathological counterexample that highlights the weakness of that algorithm. By nonpathological, I mean a natural inference problem that one may actually encounter. Here, in Appendix A, you build a rather tricky target and proposal that are really aimed at making the algorithm fail but feel rather unnatural. If I understand well, you start from a generic pi and q, and then isolate a particular pair theta,theta', and then build a new pi and a new q on an artificial state space that depend on that pair of theta values. Can standard models and proposals actually appear though this construction? * How do approximate subsampling algorithms that come with TV guarantees fit in Theorem 1? For example, the algorith MHSubLhd in your reference [6], which seems to guarantee that the algorithm, though inexact, has a limiting distribution that is within small TV distance of the original target? I fail to see what the proof of Theorem 1 says on that kind of inexact algorithm. * L136: methods based on PDMPs are dismissed maybe too quickly and harshly. Can you elaborate on your argument that the regularity conditions of refs [7, 8] only allow logistic regression? It is true that logistic regression is often used as an example, but you seem to argue that it is unable to address more examples. Although not PDMP-based, I would also add a reference here to [Pollock et al. 2020, Quasi-stationary Monte Carlo and the ScaLE Algorithm]. * Theorem 2: can you derive a corollary on how TunaMH inherits a CLT from the underlying MH, and comment on the influence of chi on the asymptotic variance? * Like Theorem 1, the actual impact of Theorem 3 is not clear to me. One of your strong points is that many existing algorithms "only" apply to easy cases like logistic regression, but one could make the opposite point that your theoretical arguments are based on adhoc "worst case" constructions of pi and q that may not be representative of any actual inference setting. However, I admit that TCS-type results such as Theorem 3 are new in the subsampling literature and definitely valuable to the general discussion. Moreover, the experiments clearly indicate the interest of the proposed algorithm, so this is a sign of support in favour of looking at worst cases to design an algorithm. * L192: I'm not sure I would call that "asymptotic optimality". I would expect "asymptotic" here to relate to N>>1. In what sense are asymptotics in M and C important here? * footnote of p7: can you support your claim that the control variates in ref 10 work only in small dimension? Relatedly, how would the panels of Figure 2 changed if I took d=10 or 50? * L374: you should formally define what an algorithm A is: what are its inputs and outputs, and what does it represent? One MCMC step? A full MCMC sweep? In particular, are theta and theta' inputs of A, or is it only the value of the ratio of proposals that matters? This will make Equation (4) clearer. * Relatedly, what is the expectation between L425 and L426 over? Same for L433? The proposed (k,z), the minibatch? This ambiguity results from the lack of a clear definition for A. * Out of curiosity, why TunaMH? I see the pun with Poisson, but not the reason for the particular choice of the tuna fish, other than the ability to "tune" chi, maybe? This is a low priority question: feel free not to answer it in the response if you don't have space. # Minor comments * L16 Maybe mention that you assume conditional independence of the data given theta. * L200 bound -> bounded * Figure 2 is too small * L391: maybe insist here that theta and theta' are held fixed throughout this section, chosen so that (4) holds. * Appendix B: maybe recall quickly the FMH and SMH Markov kernels here. * L564: strictly speaking, one needs more than detailed balance to speak of the stationary distribution pi, e.g. irreducibility.


Review 4

Summary and Contributions: The authors propose a new MH algorithm based on minibtaches. In the first part of the paper, the authors present a very nice brief review/overview of the other methods and present important theoretical results.

Strengths: The paper is well-written, clear and contains very important material.

Weaknesses: I like the paper, I have enjoyed the read. I have just some minor comments. -The choice of \xi as hyperparameter of TunaMH is quite "ugly", as you can observe in Table 2 (it makes the table a bit confused, unclear). -The proposed scheme maybe would require more explanation and more details also with some simple toy examples (for increasing the clarity of the exposition). - A discussion regarding a possible generalization with a different distribution for the size of the batches could be discussed.

Correctness: The paper seems correct and technically sound.

Clarity: Yes, in my opinion.

Relation to Prior Work: yes.

Reproducibility: No

Additional Feedback: The paper is quite complete in my opinion. It is a very good work.


Review 5

Summary and Contributions: The paper presents a mini-batching MCMC algorithm whose spectral gap is at most a constant times smaller than that of standard MH. Moreover, the tradeoff between the spectral gap and the batch size can be tuned via a single parameter. The paper includes several theorems and experimental findings in support of their approach.

Strengths: The paper presents a subsampling approach that does not require factorising the acceptance probability, which ensures a well-behaved acceptance probability and hence good ergodic behaviour. The algorithm also does not seem to require the target to concentrate in order to obtain performance benefits over standard MH, unlike various subsampling methods proposed to-date, including the Zig-Zag sampler and SMH. The paper contains multiple theoretical and experimental results in support of their approach.

Weaknesses: The upper bounds required by TunaMH (Assumption 1) are not as innocuous as is suggested. In particular, the authors describe Assumption 1 as a "local bound", and therefore not as stringent e.g. as required by Firefly. However, the bounds in Assumption 1 directly give rise to global bounds of exactly the same sort as required by Firefly. In particular, fixing an arbitrary reference point theta_0, the triangle inequality gives |U_i(theta)| <= c_i M(theta, theta_0) + |U_i(theta_0)|, which is a "global bound" in theta of exactly the same form as is used in Firefly. On the other hand, Assumption 1 is certainly weaker than the uniform upper bounds required by PoissonMH, and of the same form as used in SMH, so does not in itself necessarily weaken the standing of TunaMH compared with other related literature. However, these bounds are still a significant requirement, and this should be made more clear in the paper.

Correctness: The theorems appear to be correct, and the experiments largely support the claims made the paper. However, the code provided includes only a minimal implementation of TunaMH and MH on robust linear regression, and so does not easily permit reproducing the remaining experiments reported, including for the other methods considered. However, it would have been interesting to see the robust linear regression example reproduced in a lower-dimensional regime also, where SMH and FlyMC might be expected to perform better. This would provide further weight to the claim made that these methods break down in higher dimensions, and that TunaMH on the other hand does not.

Clarity: The paper is mostly easy to follow but would benefit from more intuition about the construction of the method. In particular, the current account given in Appendix C essentially presents TunaMH as special case of PoissonMH, i.e. an auxiliary variable method using Metropolis-within-Gibbs. However, the "factors" used (i.e. phi_i) depend on both the current point and the proposal. (The notation here in itself is misleading since e.g. phi_i(theta) depends both on theta and theta'.) While the proof of reversibility of the resulting method does seem correct, it is a bit mysterious as to what exactly is going on.

Relation to Prior Work: The contributions of this work over related literature are mostly clear. However, the authors do not necessarily provide an even-handed account of the relationship of their work to earlier methods. For example, as stated above, the bounds used in Assumption 1 are somewhat misleadingly presented as being weaker than e.g. in Firefly. Likewise, in various places the authors refer to SMH when it would be more accurate to refer to FMH. In particular, in Cornish et al. (2019), the SMH kernel is specifically designed always to use control variates - without these, the underlying kernel is equivalent to the FMH kernel used by Banterle et al. (2015) and others. The authors have made this distinction in the proof of Statement 1, which correctly refers to FMH and TFMH rather than SMH, but should do so consistently throughout the paper, including in Sections 2.2 and in the experiments. Additionally, the "Form minibatch" steps from Algorithm 2 - and in particular the part of the algorithm that decides to "eject" points from each minibatch (~lines 149-150) - are very closely related to the subsampling method used in SMH and should be attributed as such. In effect, both TunaMH and SMH sample a discrete Poisson point process process on the integers {1, ..., N} by thinning one whose intensity depends on the bounds in Assumption 1. In Appendix C, the authors cite De Sa et al., (2018) and Zhang et al. (2019) as the basis for this approach. However, while these works do also use a discrete Poisson point process, they do not involve a thinning step.

Reproducibility: Yes

Additional Feedback: The following various small points may be useful: - Eq (4) in Appendix A: The definition of curly A in Theorem 1 should be explicitly defined to return 0 or 1. At present, it seems to refer to the output of SubsMH, but SubsMH itself seems to return 'accept' (or perhaps 'reject') in Algorithm 1, and hence would not have an expectation defined - L528 and 536: "as the follows" - L545: "resamplng" - L571: The right-hand side of the equality is missing the dependence on B - L575: Again missing a dependence on B

[Author Response · NeurIPS 2020]

We thank all five reviewers for their detailed and incisive feedback. We respond in order below:

**R1: Comparison with inexact methods** Aligning with prior exact papers [10, 18], we focus on comparisons with exact
methods. We tested AustereMH [16], an inexact method, on robust linear regression in Section 5.1 with $N = 5000$. We
computed the MSE between estimated and true parameters. MH, TunaMH and AustereMH obtained MSE 0.149, 0.15
and 1.19 respectively, indicating inexact method error can be large on typical problems. We added this to the Appendix.

**R1: More difficult problem** Our next step is to apply TunaMH to more difficult problems, such as a threshold testing
with a large real-world dataset [Pierson, et al. Fast Threshold Tests for Detecting Discrimination, AISTATS 2018.]

**R1: Confusion about** $M(\theta, \theta')$ We added additional explanation here.

**R2 : What if E[B] > N** $\mathbf{E}[B]$ is typically $<< N$, and can be decreased using small step sizes. If $\mathbf{E}[B] > N$, we can
simply use standard MH in that iteration, similar to [10]. This does not affect the properties of TunaMH. We added this.

**R3: Theorem 1** (1) We believe the example in Theorem 1 is natural. The target (marginal) distribution is uniform,
the proposal is a random walk, and the state space is integers on $[0, \ldots, K - 1]$ (If $X = 0$, the example can just be
a random walk of $k$ without augmenting $z$). This is one of the simplest types of Markov chains and is encountered
in many real applications. (2) Theorem 1 is compatible with the TV bound of MHSubLhd (Proposition 3.2 in [6]).
Proposition 3.2 assumes $P_{\mathrm{MH}}$ has a bounded mixing time; it is well-known that this produces a TV bound for any kernel
by coupling [17]. Our theorem doesn't have this assumption; it suggests that for MHSubLhd with given user-specified
error, there exists a target distribution and proposal satisfying Theorem 1, on which $P_{\mathrm{MH}}$ either does not have bounded
mixing time or the mixing time is large enough such that the TV bound is greater than $\delta$. We added this to the Appendix.

**R3: PDMPs; Examples of** $U$**,** $C$**,** $M$ We have (1) weakened the language and added the suggested reference; (2) added
an example for logistic regression.

**R3: Theorem 2** Given the spectral gap bound, we can immediately show that TunaMH inherits geometric ergodicity
from MH and obtain an asymptotic variance bound — a known result [17]. In fact, the spectral gap bound and the
asymptotic variance bound are equivalent, so the impact of $\chi$ on the variance is similar to that on the spectral gap. We
unfortunately do not understand the suggestion for the corollary.

**R3: Theorem 3 impact** Similar to Theorem 1, we constructed a random walk example over two states — a simple and
natural problem. The impact is 3-fold: it (1) provides an upper bound on performance for algorithms of Algorithm 1's
form; (2) shows the upper bound is achievable (e.g. TunaMH); (3) suggests directions for developing new algorithms.
To be significantly faster than TunaMH, we either need more assumptions about the problem or new stateful algorithms.

**R3: Asymptotics in** $M$ **and** $C$ In Theorem 3 $M$ and $C$ are the only problem parameters; these two values determine a
problem's lowest possible batch size. Thus, asymptotic optimality in $M$ and $C$ indicates asymptotic optimality in any
problem parameter (including $N$). When explicitly assuming the relation of $N$ and $M$, $C$ (as in prior work [10]), in
Corollary 1 we show how the bound depends on $N$ and TunaMH is asymptotically optimal in $N$.

**R3: Claim about control variates** We empirically observe SMH with control variates does not work well in high
dimension. We conjecture this is because the batch size bound (Eq.13 in [10]) becomes looser quickly as $d$ increases.

**R3&R5: Results in low dimension** We ran experiments (Section 5.1) with $d = 10$ and $N = 10^5$. The ESS/S for SMH,
FlyMC, TunaMH are 0.02, 0.75, & 1.7; SMH-1, SMH-2, FlyMC-MAP and TunaMH-MAP are 174.7, 5969.5, 730.8, &
730.1 respectively. This suggests TunaMH is significantly better without MAP/control variates. With MAP/control
variates, TunaMH is better than SMH-1, similar to FlyMC and worse than SMH-2. We will add this to the Appendix.

**R3:** $\mathcal{A}$ **in Appendix** $\mathcal{A}$ denotes the `SubsMH` of the minibatch MH method in question. The expectation in L425, L426
and L433 are all taken over the randomness in `SubsMH`. We have clarified this.

**R4: Table 2; other distributions for batch size** (1) We fixed Table 2 to make $\chi$ values clearer. (2) We were unable to
identify a distribution other than Poisson to get nice results. It is possible one does exist, which we leave to future work.

**R5: Assumption 1** For consistency we use the same language as in prior work [10] to describe the bounds in Assumption
1 and in FlyMC. One can get the required bound in FlyMC by triangle inequality, but the tightness of the bound highly
depends on the reference point $\theta_0$. Thus it is typically harder to get a tight bound on $|U_i(\theta)|$ than $|U_i(\theta) - U_i(\theta')|$. We
agreed that Assumption 1 is still a strong requirement. We have clarified these points in the paper.

**R5: Reproducibility; Clarity about SMH and FMH** We (1) will release the code and associated documentation upon
publication; (2) have cleaned up our SMH/FMH language to be more precise.

**R5: "Form minibatch" steps** Thinning is a well-known technique (Lewis and Shedler, 1979) used in many papers [7,
8, 24]. We developed the "Form minibatch" steps, including ejection, by replacing the global bounds with the local
bounds in Algorithm 4 in the Appendix of [24] (explained in Appendix C). We clarified this and now also cite [10].

[Meta-Review · NeurIPS 2020]

This paper presents a new minibatch MH algorithm. All the reviewers participated to the discussion after the rebuttal was made available and were unanimously positive about it. * Strengths - The algorithm preserves the correct invariant distribution and shows that algorithms which do not preserve it can go seriously wrong. - The authors provide an interesting theoretical analysis of the algorithm. * Weakenesses - The paper would benefit from a more intuitive introduction of Tuna-MH - A limitation of previous subsampling MH type algorithms is that they appear to be only beneficial in scenarios where the posterior concentrates and is approximately Gaussian as emphasized in Bardenet et al. 2017; Cornish et al., 2019. It should be clarified in the paper whether this is also the case for the method presented here. - The PDMP methods are not only applicable to logistic regression as claimed by the authors.